



# Calculation of Uncertainty in the (U-Th)/He System

Peter E. Martin[1], James R. Metcalf[1], Rebecca M. Flowers[1]

[1]Department of Geological Sciences, University of Colorado Boulder, CO 80309

*Correspondence to*: Peter E. Martin (Peter.Martin-2@colorado.edu)

**Abstract.** Currently there is no standardized approach for reporting date uncertainty in the (U-Th)/He system, partly due to the fact that the methods and formulae for calculating single-grain uncertainty have never been published. This creates challenges for interpreting the expected distribution of dates within individual samples and of dates generated by differing labs. Here we publish two procedures to derive (U-Th)/He single-grain date uncertainty (linear and Monte Carlo uncertainty propagation), based on input $^4$He, radionuclide, and isotope-specific $F_T$ values and uncertainties. We also describe a newly released software package, HeCalc, that performs date calculation and uncertainty propagation for (U-Th)/He data. Using this software, we find that date uncertainty decreases with increasing age for constant relative input uncertainties. Skew in date probability distributions (i.e., asymmetrical uncertainty) yielded by the Monte Carlo method varies with age and increases with increasing relative input uncertainty. Propagating uncertainties in $^4$He and radionuclides using a compilation of real (U-Th)/He data (N = 1978 apatites, and 1753 zircons) reveals that the uncertainty budget in this dataset is dominated by uncertainty stemming from the radionuclides, yielding median relative uncertainty values of 2.9% for apatites and 1.7% for zircons. When uncertainties in $F_T$ of 2% or 5% are assumed and additionally propagated, these values increase to 3.3% and 5.0% for apatite, and 2.4% and 4.7% for zircon. The potentially strong influence of $F_T$ on the uncertainty budget indicates the need to better quantify and routinely propagate $F_T$ uncertainty into (U-Th)/He dates. Skew is generally positive and can be significant, with ~14% of apatite dates and ~5% of zircon dates characterized by skew of 10% or greater. This outcome indicates the value of applying Monte Carlo uncertainty propagation to identify samples with substantially skewed uncertainties that should be considered during data interpretation. The formulae published here and the associated HeCalc software can aid in more consistent (U-Th)/He uncertainty reporting and enable a more rigorous understanding of when and why multiple aliquots from a sample differ beyond what is expected from analytical and $F_T$ uncertainties.

## 1 Introduction

Geochronology and thermochronology by the (U-Th)/He method was initially developed as a reliable technique approximately three decades ago (Farley et al., 1996; Wernicke and Lippolt, 1994; Wolf et al., 1996; Zeitler et al., 1987). Since that time, numerous advances such as the ability to measure the (U-Th)/He date of individual grains (e.g., House et al., 2000), improvements in kinetic models to account for the effects of radiation damage and annealing on He diffusion kinetics (e.g., Flowers et al., 2009; Gautheron et al., 2009; Guenthner et al., 2013), and the development of thermal history modeling tools



that improve interpretation of these data (Gallagher, 2012; Ketcham, 2005) have led to the widespread application of this technique and large amounts of data generation. However, with this progress has come recognition of the need to more rigorously and consistently report uncertainties on individual (U-Th)/He dates (Flowers et al., 2022a; Ketcham et al., 2022). For example, the intra-sample variability of (U-Th)/He dates often exceeds that predicted by analytical uncertainty, due both to interpretable variation from kinetic differences among grains of the same sample and to uninterpretable scatter from other

factors (e.g., Brown et al., 2013; Fitzgerald et al., 2006; Flowers et al., 2022a; Flowers and Kelley, 2011). Better accounting for the uncertainties of individual analyses is a key first step in determining whether multiple individual analyses from a sample are actually "over-dispersed", and would help develop a more complete understanding of the causes of data dispersion (Flowers et al., 2022b). More rigorous uncertainty reporting would also improve confidence in large-N datasets, facilitate inter-laboratory data comparisons, and ultimately increase the precision and accuracy of thermal history reconstructions.

40        One current challenge to comprehensive uncertainty propagation is that the methods for propagating uncertainty components into single-grain (U-Th)/He dates have never been described in the literature, and the formal analytical uncertainty in (U-Th)/He dates has never been thoroughly assessed. Uncertainty propagation in the (U-Th)/He system is complicated by the fact that the age equation has no analytical solution, precluding the direct application of typical specific uncertainty propagation formulas that combine individual uncertainty components in quadrature through a given function. This problem

may be circumvented by approximations of the He age equation that solve directly for time (e.g., Meesters and Dunai, 2005), or by the use of the general "error propagation equation" using the first derivatives of the uncertainty components with respect to time (Bevington and Robinson, 2003). However, linear uncertainty propagation methods rely on an assumption that the derivative of the first term of the Taylor series is a linear function at the scale of the uncertainties being combined (Bevington and Robinson, 2003; McLean et al., 2011). As this assumption is often violated in the (U-Th)/He system, uncertainties have the potential to be skewed (i.e., asymmetrical), and uncertainties propagated using standard linear uncertainty propagation may

be inaccurate.

        Comprehensive uncertainty accounting on individual (U-Th)/He dates involves propagating not only the analytical uncertainties associated with measurements of parent and daughter amounts, but also propagation of uncertainties associated with alpha-ejection corrections ($F_T$ corrections, which account for He ejected from the crystal via alpha decay). While the

analytical uncertainty on parent and daughter amounts is generally well-characterized, quantifying the uncertainty in $F_T$ values for various minerals and grain geometries remains an active area of research (e.g., Cooperdock et al., 2019; Zeigler et al., 2021). As $F_T$ uncertainties are better quantified, propagating both analytical and $F_T$ uncertainties into the reported uncertainty of (U-Th)/He dates is desirable (e.g., Flowers et al., 2022b).

        Here we describe in detail how analytical and $F_T$ uncertainties in (U-Th)/He dates may be combined to derive a single-

grain (U-Th)/He date uncertainty. To address the shortcomings of linear uncertainty propagation, we primarily adopt a Monte Carlo approach to quantitatively constrain (U-Th)/He uncertainty. This procedure is both accurate and mathematically simple, and enables evaluation of asymmetric uncertainties (which linear uncertainty propagation does not provide). For completeness and to ease retrospective data comparisons, we also include a method to propagate uncertainty through the calculation of a (U-



Th)/He date that relies on more traditional linear uncertainty propagation. In addition, this manuscript presents a new program
written in Python 3.8 termed HeCalc (Helium date and uncertainty Calculator; Martin, 2022) that is capable of performing
both Monte Carlo and linear methods of uncertainty propagation. Using this new software, we apply these uncertainty
propagation methods to a sensitivity analysis of the overall behavior of (U-Th)/He uncertainty as a function of the various
input uncertainties and the resulting date. We also compare the results from linear and Monte Carlo methods to examine the
potential limitations resulting from inaccuracy of linear uncertainty propagation. Finally, we use HeCalc to reduce a
compilation of real data to determine the typical contributions of each uncertainty component to date uncertainty in actual
practice.

## 2 Background: uncertainty components in (U-Th)/He dates

The currently quantifiable uncertainties on single-grain (U-Th)/He dates include analytical uncertainties associated
with parent and daughter isotope measurements and geometric uncertainties associated with alpha-ejection corrections. These
are discussed in detail in Flowers et al. (2022a) and summarized more briefly here. We use the word "uncertainty" as a
probabilistic statement of the distribution of repeated measurements (e.g., for $^{238}$U of $10 \pm 1$ ppm at $1\sigma$, 68.27% of repeated
measurements will fall between 9 and 11 ppm), while "error" refers to deviation of a measured value from the true value.

In the (U-Th)/He technique, the parent nuclides (Uranium, Thorium, and Samarium; $^{238}$U, $^{235}$U, $^{232}$Th, $^{147}$Sm) are
typically measured using inductively coupled plasma mass spectrometry (ICP-MS), while the daughter product (Helium; $^{4}$He)
is usually measured on a dedicated quadrupole or sector noble gas mass spectrometer. Most commonly, quantification of
helium and its parent nuclides is performed via isotope spike to permit conversion from ratioed mass spectrometric
measurements to molar amounts. Given the measurements of parent and daughter products, a (U-Th)/He date may be calculated
using the equation for $^{4}$He ingrowth

$$^{4}He= 8\,^{238}U\left(e^{\lambda_{238}t}-1\right)+7\,^{235}U\left(e^{\lambda_{235}t}-1\right)+6\,^{232}Th\left(e^{\lambda_{232}t}-1\right)+^{147}Sm\left(e^{\lambda_{147}t}-1\right)$$

(1)

where each nuclide is given as an amount, $t$ is time, and $\lambda$ is the decay constant for each parent nuclide given in the subscript.

Because of the kinetic energy associated with alpha decay, individual alpha particles (i.e., $^{4}$He nuclei) travel between
4 and 34 µm in solid matter, depending on parent nuclide, before coming to rest (Farley et al., 1996; Ketcham et al., 2011).
This redistribution of the daughter product can result in daughter helium being ejected from a crystal. By assuming a
homogenous parent nuclide distribution, measuring the physical dimensions of a single grain, and applying a geometric model
to those physical dimensions, the proportion of alpha particles retained in a grain (the fraction trapped; $F_T$) can be calculated
for each nuclide's mean stopping distance (Ketcham et al., 2011). Determination of grain dimensions to calculate $F_T$ is usually
accomplished via size measurement of individual grains using photomicrographs with a calibrated digital camera (Cooperdock





et al., 2019; Glotzbach et al., 2019). Using the $F_T$ parameter, the effects of alpha ejection on a date can be corrected using a
modified version of the $^4$He ingrowth equation (Ketcham et al., 2011):

$$^4\text{He}= 8^{238}F_T^{238}U\left(e^{\lambda_{238}t}-1\right)+7^{235}F_T^{235}U\left(e^{\lambda_{235}t}-1\right)+6^{232}F_T^{232}\text{Th}\left(e^{\lambda_{232}t}-1\right)+^{147}F_T^{147}\text{Sm}\left(e^{\lambda_{147}t}-1\right)$$

(2)

We refer to dates calculated with this correction applied as "alpha ejection corrected" or simply "corrected" dates, while dates
calculated with no correction applied using Eq. (1) we refer to as "uncorrected" or "raw" dates.

100        For this work, it is assumed that the amount and uncertainty of each nuclide has been constrained. The natural U
isotope ratio (137.818 ± 0.023 1s; Hiess et al., 2012) is usually used to calculate $^{235}$U in a sample based on the $^{238}$U amount
measured. In these cases, the uncertainty in $^{235}$U is perfectly correlated with $^{238}$U; treatment of these uncertainties as though
they were independent could lead to inaccurate uncertainty calculations. Correlated uncertainty between other nuclides is likely
negligible, depending on the exact procedure used for isotope spiking. As these calculations do not typically involve common
isotopes in multiple ratios (e.g. the correlated uncertainty resulting from the measurement of $^{206}$Pb/$^{204}$Pb and $^{207}$Pb/$^{204}$Pb in Pb-
Pb dating; McLean et al., 2011), the radionuclide uncertainty correlations in (U-Th)/He dating will be related only to systematic
error introduced as a result of adding a common spike solution. The most common method of adding spike involves pipetting;
precision for a typical modern pipette is such that other random error will overwhelm this uncertainty contribution. The option
to incorporate correlated radionuclide uncertainty is included in the methods for propagating uncertainty, but we make the
simplifying assumption in the discussion that these uncertainties are fully uncorrelated.

        Error in $F_T$ values likely stems from a combination of undetected parent nuclide zonation (Farley et al., 1996),
inaccurate size measurements, and assumptions regarding the specific geometry of a given grain (i.e., deviations from the
"idealized" shapes in Ketcham et al., 2011); when the magnitude of these effects is constrained they can be propagated into $F_T$
value uncertainty. Measurement of parent nuclide zonation is not currently possible in typical workflows, so this source of
error is generally unquantified for routine analyses. Several approaches have been developed to approximate the three-
dimensional shape of individual grains to assess uncertainty associated with 2D grain measurement, generally finding that
uncertainty due to errors in grain geometry measurement range between 2-9% (Cooperdock et al., 2019; Evans et al., 2008;
Glotzbach et al., 2019; Herman et al., 2007; Zeigler et al., 2021), though these methods are yet routinely included in workflows.
These initial studies suggest that uncertainty in $F_T$ may be significant relative to uncertainty from mass spectrometric
measurements (which are also often in the range of a few percent; Sect. 5.4). While standardized and straightforward methods
of constraining the uncertainty in $F_T$ do not currently exist, mathematically including uncertainty in $F_T$ values for date
uncertainty calculations is possible and efforts are underway to permit incorporation of generalized $F_T$ geometric uncertainty
values for a wide range of grain shapes in routine analyses (Zeigler et al., 2021). We therefore include uncertainty on $F_T$ in the
following methods of (U-Th)/He date uncertainty propagation, including only estimates of geometric uncertainty in these
initial analyses. Future measurements of uncertainty owing to parent nuclide zonation could also be included in overall $F_T$
uncertainty if labs characterize zonation prior to date measurement. The potential causes of variance (parent nuclide zonation,
errors in size measurement, non-ideal grain shapes) will be shared between isotope-specific $F_T$ values and likely result in highly



correlated uncertainty. No study has examined the extent of covariance in $F_T$ uncertainties, so here we include correlated $F_T$ uncertainty in the methods below and perform uncertainty propagation with both fully correlated and fully uncorrelated $F_T$ uncertainty in the discussion.

Several additional sources of variance in (U-Th)/He dates exist that we do not include in uncertainty propagation here because they are generally not possible to quantify or are not routinely measured (e.g., Flowers et al., 2022b, 2022a). In contrast with the above sources of uncertainty, these factors potentially contribute to intra-sample variability, but would not cause variance in repeated measurements of the same grain, indicating that these sources are best considered as part of multi-aliquot data compilations. These include parent nuclide zonation (e.g., Farley et al., 2011; Hourigan et al., 2005), alpha implantation (Murray et al., 2014), and deviations from expected diffusion behavior (e.g., Zeitler et al., 2017). The uncertainty in decay constants is negligible relative to other sources of uncertainty and results in systematic error across all (U-Th)/He measurements, and therefore is not incorporated in our uncertainty calculation methods.

## 3 Date and uncertainty calculation methods

Here, (U-Th)/He dates are calculated by first estimating a date using an approximation of the helium age equation that solves directly for time. Using this estimate as an initial value, the exact date is then calculated iteratively using the Newton-Raphson method. We describe two independent methods (linear uncertainty propagation and Monte Carlo uncertainty modeling) of calculating the uncertainty in this date given the uncertainty components described in Sect. 2 above. We exclusively use the term "linear uncertainty propagation" rather than "analytical" or "standard" propagation to avoid confusion with analytical error arising from instrument noise and standards used in analytical measurements, respectively. As discussed in detail in Sect. 5.3 below, the linear method allows precise and repeatable calculations, while the Monte Carlo method is slightly more accurate.

### 3.1 Date calculation

The initial value for iterative age calculation is obtained by calculating an approximated noniterative solution of the (U-Th)/He age equation as described by Meesters and Dunai (2005). We slightly modify this method to permit calculation of parent-specific alpha ejection-corrected effective helium production rates:

$$p_j = N \times {}^jF_T \times \lambda_j \times {}^jM$$

(3)

Where $p_j$ is the $^4$He production rate, N is the number of alpha particles produced by a given decay chain, and ${}^jF_T$, $\lambda_j$, and ${}^jM$ are the alpha ejection-correction factor, decay constant, and concentration of radionuclide $j$ (i.e. $^{238}$U, $^{235}$U, $^{232}$Th, and $^{147}$Sm), respectively. As the $^{235}$U amount is generally presumed to be 1/137.818 that of $^{238}$U, a further modification can be made in this case:



$$p_{235} = 7 \times {}^{235}F_T \times \lambda_{235} \times \frac{{}^{238}U}{137.818}$$

(4)

Following these modifications, the approximate date may be calculated by first computing the total alpha ejection-corrected production rate ($P$) and a mean decay constant weighted by effective production rate ($\lambda_{wm}$):

$$P = \sum_{j=1}^{4} p_j$$

(5)

$$\lambda_{wm} = \sum_{j=1}^{4} \frac{p_j}{P} \lambda_j$$

165  (6)

$$t = \frac{1}{\lambda_{wm}} \ln\left(\frac{\lambda_{wm} \times [He]}{P} + 1\right)$$

(7)

Using this approximation as an initial guess ($t_0$), the (U-Th)/He date is then found using the relatively simple but highly efficient Newton-Raphson method


$$t_{i+1} = t_i - \frac{f(t_i)}{f'(t_i)}$$

(8)

$$f(t_i) = 0 = \left[\sum_{j=1}^{4} N \, {}^{j}F_T \, {}^{j}M\left(e^{\lambda_j t_i} - 1\right)\right] - He$$

(9)

$$t_{i+1} = t_i - \frac{\left[\sum_{j=1}^{4} N \, {}^{j}F_T \, {}^{j}M\left(e^{\lambda_j t_i} - 1\right)\right] - He}{\sum_{j=1}^{4} N\lambda_j \, {}^{j}F_T \, {}^{j}M e^{\lambda_j t_i}}$$

175  (10)

where $t_i$ and $t_{i+1}$ are successive approximations of the date, and $f(t_i)$ and $f'(t_i)$ are the implicit age equation (the helium age equation set at zero; Eq. (9)) and its first derivative with respect to $t$, respectively. This calculation is repeated until the difference between successive iterations is less than one year. This method benefits from an accurate initial guess and a quadratic rate of convergence such that generally only three to five iterations are required, though for dates >500 Ma (where



the noniterative approximation produces relative errors of >0.1% ; Meesters and Dunai, 2005), as many as ten iterations may
be required.

## 3.2 Linear uncertainty propagation

Here we provide a method of calculating date uncertainty using linear propagation of uncertainty. We apply the
general formula for uncertainty propagation through a function $f(a, b…z)$, including cross terms for correlated error where
such correlations exist (Bevington and Robinson, 2003):

$$\sigma_f = \sqrt{\left(\frac{\partial f}{\partial a}\sigma_a\right)^2 + \left(\frac{\partial f}{\partial b}\sigma_b\right)^2 + 2\frac{\partial f}{\partial a}\frac{\partial f}{\partial b}\sigma_{ab}{}^2 + \cdots + \left(\frac{\partial f}{\partial z}\sigma_z\right)^2}$$

(11)

The following equations presume that $^{235}U$ has not been measured directly, but equations that include directly
quantified $^{235}U$ are provided in the appendix, and the HeCalc software released with this paper includes an option to account
for either means of constraining $^{235}U$. As an alternative to the use of HeCalc, these equations could be replicated in spreadsheet
programs with a one-time expenditure of effort.

Applying the uncertainty propagation equation to the (U-Th)/He age equation, including potential covariance in the
radionuclide and $F_T$ uncertainties (i.e., the potential that the uncertainties are not fully independent), indicates that the
uncertainty in a (U-Th)/He date is:

$$\sigma_t = \left[ \begin{aligned} &\left(\frac{\partial t}{\partial\,^4He}\sigma_{He}\right)^2 + \left(\frac{\partial t}{\partial\,^{238}U}\sigma_{238}\right)^2 + \left(\frac{\partial t}{\partial\,^{232}Th}\sigma_{232}\right)^2 + \left(\frac{\partial t}{\partial\,^{147}Sm}\sigma_{147}\right)^2 + \\ &2\frac{\partial t}{\partial\,^{238}U}\frac{\partial t}{\partial\,^{232}Th}\sigma_{238-232}{}^2 + 2\frac{\partial t}{\partial\,^{238}U}\frac{\partial t}{\partial\,^{147}Sm}\sigma_{238-147}{}^2 + 2\frac{\partial t}{\partial\,^{232}Th}\frac{\partial t}{\partial\,^{147}Sm}\sigma_{232-147}{}^2 + \\ &\left(\frac{\partial t}{\partial\,^{238}F_T}\sigma_{Ft238}\right)^2 + \left(\frac{\partial t}{\partial\,^{235}F_T}\sigma_{Ft235}\right)^2 + \left(\frac{\partial t}{\partial\,^{232}F_T}\sigma_{Ft232}\right)^2 + \left(\frac{\partial t}{\partial\,^{147}F_T}\sigma_{Ft147}\right)^2 + \\ &2\frac{\partial t}{\partial\,^{238}F_T}\frac{\partial t}{\partial\,^{235}F_T}\sigma_{Ft238-Ft235}{}^2 + 2\frac{\partial t}{\partial\,^{238}F_T}\frac{\partial t}{\partial\,^{232}F_T}\sigma_{Ft238-Ft232}{}^2 + \\ &2\frac{\partial t}{\partial\,^{238}F_T}\frac{\partial t}{\partial\,^{147}F_T}\sigma_{Ft238-Ft147}{}^2 + 2\frac{\partial t}{\partial\,^{235}F_T}\frac{\partial t}{\partial\,^{232}F_T}\sigma_{Ft235-Ft232}{}^2 + \\ &2\frac{\partial t}{\partial\,^{235}F_T}\frac{\partial t}{\partial\,^{147}F_T}\sigma_{Ft235-Ft147}{}^2 + 2\frac{\partial t}{\partial\,^{232}F_T}\frac{\partial t}{\partial\,^{147}F_T}\sigma_{Ft232-Ft147}{}^2 \end{aligned} \right]^{1/2}$$

(12)

where, for example, $\sigma_{He}$ is the uncertainty in the $^4He$ measurement, and $\sigma_{238-232}$ is the covariance between $^{238}U$ and $^{232}Th$. Note
that the covariance terms collapse to 0 if no correlation exists between uncertainties, while positive covariance will increase
the overall uncertainty.



While solving the (U-Th)/He age equation for $t$ explicitly is not possible, finding the first derivative of $t$ with respect to each variable is possible through implicit differentiation. Specifically,

$$\frac{\partial t}{\partial X} = -\frac{\frac{\partial f}{\partial X}}{\frac{\partial f}{\partial t}}$$

(13)

where $X$ is each variable in the (U-Th)/He age equation with an uncertainty. Using this relationship, the relevant derivatives

are:

$$\frac{\partial f}{\partial\ ^4He} = \frac{1}{\sum_{j=1}^{4} N\lambda_j\ ^jF_T\ ^jM e^{\lambda_j t_i}}$$

(14)

$$\frac{\partial f}{\partial\ ^{238}U} = -\frac{8\ ^{238}F_T\left(e^{\lambda_{238} t_i} - 1\right) + \frac{7}{137.818}\ ^{235}F_T\left(e^{\lambda_{235} t_i} - 1\right)}{\sum_{j=1}^{4} N\lambda_j\ ^jF_T\ ^jM e^{\lambda_j t_i}}$$

(15)

$$\frac{\partial f}{\partial\ ^{232}Th} = -\frac{6\ ^{232}F_T\left(e^{\lambda_{232} t_i} - 1\right)}{\sum_{j=1}^{4} N\lambda_j\ ^jF_T\ ^jM e^{\lambda_j t_i}}$$

(16)

$$\frac{\partial f}{\partial\ ^{147}Sm} = -\frac{^{147}F_T\left(e^{\lambda_{147} t_i} - 1\right)}{\sum_{j=1}^{4} N\lambda_j\ ^jF_T\ ^jM e^{\lambda_j t_i}}$$

(17)

$$\frac{\partial f}{\partial\ ^jF_T} = -\frac{N\ ^jM\left(e^{\lambda_j t_i} - 1\right)}{\sum_{j=1}^{4} N\lambda_j\ ^jF_T\ ^jM e^{\lambda_j t_i}}$$

215    (18)

Where each summation term involves addition of the four radionuclides with the same variable convention described in Sect.

2.1.1 above and $\frac{^{238}U}{137.818}$ used in place of $^{235}U$, e.g.:

$$\sum_{j=1}^{4} N\lambda_j\ ^jF_T\ ^jM e^{\lambda_j t_i} = \begin{bmatrix} 8\lambda_{238}\ ^{238}F_T\ ^{238}U e^{\lambda_{238} t_i} + \frac{7}{137.818}\lambda_{235}\ ^{235}F_T\ ^{238}U e^{\lambda_{235} t_i} + \\ 6\lambda_{232}\ ^{232}F_T\ ^{232}Th e^{\lambda_{232} t_i} + \lambda_{147}\ ^{147}F_T\ ^{147}Sm e^{\lambda_{147} t_i} \end{bmatrix}$$



$$(19)$$

These equations are printed in their expanded forms, along with versions that allow for direct quantification of $^{235}$U, in the appendix.

### 3.3 Monte Carlo uncertainty modeling

### 3.3.1 Monte Carlo uncertainty calculations

        Monte Carlo uncertainty propagation is based on the approach of combining the uncertainty in measured parameters
with any given probability distribution by randomly sampling each distribution a large number of times and propagating those randomly generated parameters through some function of interest (Eqs. (7) and (10); Fig. 1). This method yields a probability density histogram that describes the true uncertainty to arbitrary precision depending on the number of simulations run (Anderson, 1976; Efron and Tibshirani, 1986; Possolo and Iyer, 2017). As such, the application of Monte Carlo techniques is mathematically straightforward, in this case requiring no knowledge beyond that required to calculate a (U-Th)/He date. In
addition to this benefit, Monte Carlo uncertainty analysis does not require that a function of interest have a linear first term of the Taylor series to accurately calculate uncertainty; when this assumption is violated (as in the (U-Th)/He age equation), uncertainties propagated using linear uncertainty propagation (Eq. (11)) can be inaccurate. While the Monte Carlo method has historically been hindered by computational expense, the increases in computational power in recent decades make this more accurate approach an attractive method for routine uncertainty propagation in (U-Th)/He chronology.

Here, Monte Carlo uncertainty modeling of (U-Th)/He data is performed by generating arrays of a pre-determined size N, which contain randomly generated values for each input according to the gaussian distribution described by each value's $1\sigma$ uncertainty (Fig. 1, input probability distributions). Correlated uncertainties (correlations between $^{238}$U, $^{232}$Th, and $^{147}$Sm, and also between $^{238}$F$_T$, $^{235}$F$_T$, $^{232}$F$_T$, and $^{147}$F$_T$) are generated using multivariate gaussian distributions according to a covariance matrix consisting of each value's $1\sigma$ uncertainty and the covariance term for each pair of variables. Arrays of raw
and corrected dates of size N are then calculated as described above using these randomly generated variables. From these arrays, 68% confidence intervals are calculated using the 15.865 and 84.135 percentiles of the samples of dates. We use confidence intervals as opposed to standard deviation because some output uncertainty distributions are skewed. Although the average of the 68% confidence intervals yields the standard deviation for reasonably gaussian (normal) distributions, this does not necessarily hold for non-gaussian (asymmetric or skewed) distributions (Fig. 1, example output probability distributions;
Sect. 5.2).





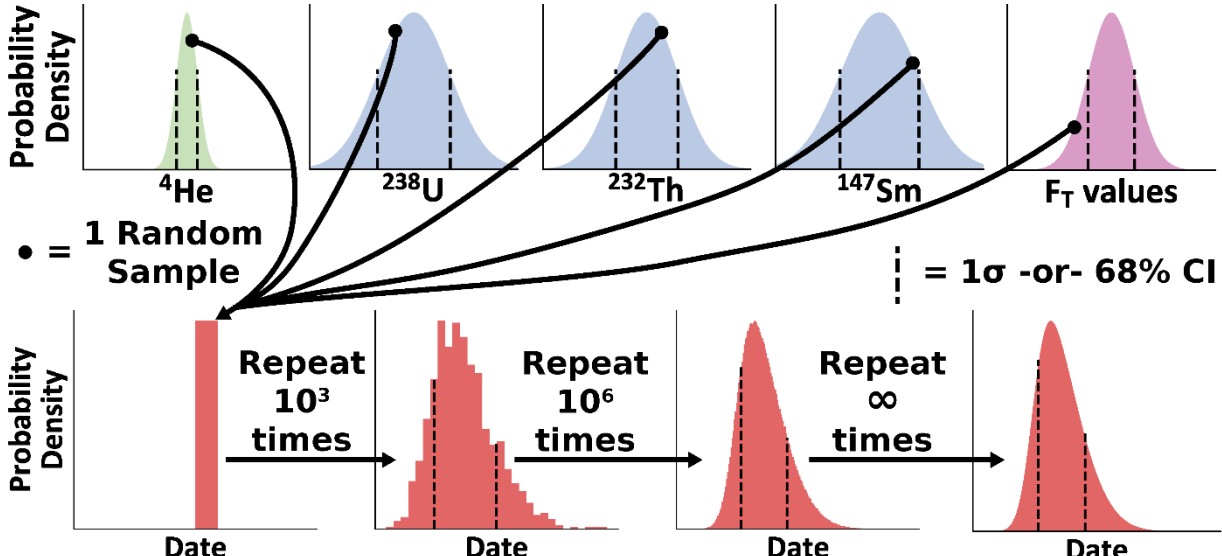

**Figure 1: A conceptual diagram of Monte Carlo uncertainty modeling for the (U-Th)/He system. Each independent gaussian input probability distribution (with 1σ standard deviation marked as vertical lines) is sampled at random a large number of times. Because we assume fully correlated F$_T$ uncertainties, the isotope-specific F$_T$ distributions are represented by a single distribution where a single percent deviation from the mean value is sampled four times. Using these randomly sampled inputs, a single date is calculated. This process is repeated until the probability distribution of interest (in this case, a skewed non-gaussian distribution with the 68% confidence interval shown with vertical lines) has been sufficiently sampled, as determined by the analyst.**

### 3.3.2 Precision of Monte Carlo method

Because Monte Carlo analysis is a numerical approximation of uncertainty, the number of simulations dictates the precision of the results (e.g., the lower panels in Fig. 1 become progressively smoother with an increasing number of simulations). Therefore, separate from the probability distribution describing date uncertainty, there is a predictable level of variation in uncertainty estimates and other parameters describing the probability distribution (e.g., its mean) given a certain number of total Monte Carlo simulations (Wübbeler et al., 2010). Specifically, the standard deviation of the mean value of a Monte Carlo model is dependent on the uncertainty in the value itself and the number of simulations:

$$\sigma_\mu = \frac{\sigma_t}{\sqrt{N}}$$

(20)

where $\sigma_\mu$ is the standard deviation of the population mean, $\sigma_t$ the date uncertainty, and $N$ the number of simulations. To avoid running arbitrary numbers of simulations, we invert this equation to determine the number of iterations required to achieve a user-requested relative precision on the mean:

$$\sigma_\mu \sim (\bar{x} \times p)$$

(21)



$$N = \frac{\sigma_t{}^2}{(\bar{x} \times p)^2}$$

(22)

Where $N$ is the number of simulations to run, $\sigma_t$ is the date uncertainty estimated by linear uncertainty propagation, $\bar{x}$ is the
270 sample mean estimated by calculation of the date using the nominal input values, and $p$ is the user-requested precision in
percent uncertainty. By using percent relative uncertainty, the value of the date itself need not be known *a priori*, as an estimate
of the standard deviation of the population mean date can be obtained on the fly using the percent relative precision and the
date calculation from the input values (Eq. (22)).

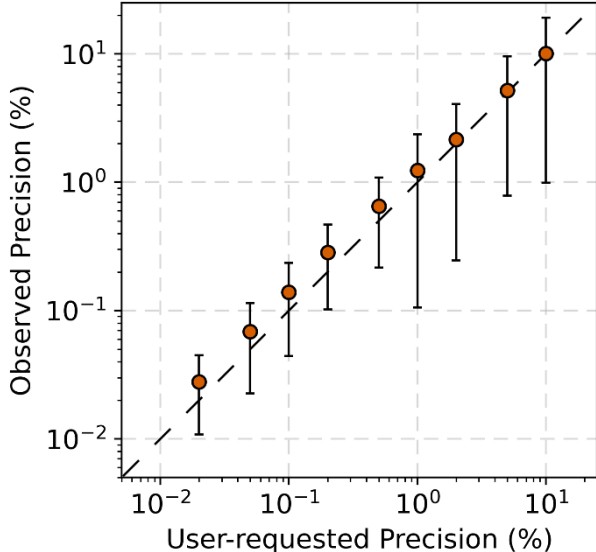

275 **Figure 2: Requested mean date precision for the Monte Carlo compared to that empirically observed by running the same model
100 times in succession for a range of input values and uncertainties. The dashed line shows a one-to-one relationship.**

This method of estimating precision was validated by running the Monte Carlo code 100 times in succession for a
range of date and relative uncertainty value permutations. By taking the standard deviation of the output mean value for the
repeated calculations, we derive the empirical precision of the estimates of mean uncertainty. Comparing these to the observed
and user-requested precisions, this method results in a strong 1-to-1 correspondence between the observed and user-requested
precision, as shown in Fig. 2.



**4 Helium date and uncertainty Calculator (HeCalc) code**

In this section we describe the implementation of the above methods of date and uncertainty calculation in the new HeCalc software (Martin, 2022). For ease of access and to best provide this software as a resource to the helium community,

HeCalc is available both as a standalone program with a graphical user interface (GUI) and as a package in Python 3 available for download from the Python Package Index (PyPI) via pip commands. The descriptions below apply specifically to the GUI version of the software and the main_hecalc() function in the Python package; those interested in writing their own code and incorporating the component functions provided in HeCalc may consult the associated documentation for more detailed programming considerations.

**4.1 Input**

The input for HeCalc is designed to be straightforward and flexible. Input files may be in Excel (.xls/.xlsx), comma separated value (.csv), or tab-delimited text (.txt) format. In addition to data input through a file, HeCalc users may manually input values to calculate a date and uncertainty for a single set of data by clicking on the "Manual" tab. If importing data through a file, the file must contain columns for sample name, U, Th, Sm, He, and all $F_T$s with the headers Sample, mol 238U,

mol 232Th, mol 147Sm, mol 4He, 238Ft, 235Ft, 232Ft, and 147Ft (Table 1). Although "mol X" is required as the input column header for U, Th, Sm, and He, the actual units of the input data may be any unit of quantity (e.g., atoms, mol/g, etc.) as long as they are identical. The $1\sigma$ uncertainty for each value, in the same units, must be included in the column following each respective value, even if the applied uncertainty is 0 (e.g., for $F_T$ values with unknown uncertainty); there is no naming requirement for these headers. If $^{235}$U was measured directly, columns for this measurement and its uncertainty should also be

present. Correlated uncertainty between the radionuclides and between the isotope-specific $F_T$ values can be input using their Pearson correlation coefficient, which is related to the covariance as:

$$r_{ab} = \frac{\sigma_{ab}{}^2}{\sigma_a \sigma_b}$$

(23)

where $\sigma_{ab}$ is the covariance between variables *a* and *b*. The correlation coefficient is preferable to inputting covariance directly

as it has the intuitive meaning of being in the range of [-1, 1] where 1 is perfectly correlated and 0 is fully uncorrelated, while numerical covariance is generally unintuitive. These values may be included in the input file using headers with the naming convention "r 238U-235U" and "r 238Ft-235Ft" (Table 1); either ordering of the correlated uncertainties in the header (i.e., "r 238U-235U" vs. "r 235U-238U") is permitted. Uncertainties are assumed to be uncorrelated unless these columns are explicitly included. Example input files both with and without correlated uncertainty are provided as templates in the code's repository

(see the Code availability Section for a direct link).

The order in which these columns appear is unimportant as long as the uncertainty associated with each value follows that value. Extraneous columns with differing headers also will not interfere with the code's execution. Additionally, if an





input Excel file has multiple sheets, the first sheet will be read in by default. If this sheet does not contain the required column headers, the program will ask for the name of the sheet to use instead. In this way, HeCalc ideally allows for input of any given

lab's standard data reduction spreadsheet or other typical data product with no or minimal alteration, allowing it to be integrated seamlessly into a lab's existing workflow.

In addition to data input, several further options are provided. The number of decimals included in the output is determined by the user (this option affects only output and does not impact the statistical aspects of the code). The user can also select whether to perform linear uncertainty propagation, Monte Carlo uncertainty propagation, both, or neither. If Monte

Carlo uncertainty propagation is selected, the desired precision of the mean is specified in percent as described above. In practice, the precision of the mean date need be no better than the number of significant figures present the in data; for common (U-Th)/He analyses, this equates to a precision of ~0.01%, which generally requires on the order of $10^4$-$10^5$ simulations. The program also contains the ability to generate histograms using the Monte Carlo results. If this option is chosen, this histogram may be parameterized as a skew-normal distribution (Azzalini and Capitanio, 1999; O'Hagan and Leonard, 1976).

| Column header | Example required data input |
|---|---|
| Sample | Sample1 |
| mol 4He | 0.1 |
| ± | 0.001 |
| mol 238U | 1 |
| ± | 0.05 |
| mol 232Th | 1 |
| ± | 0.05 |
| mol 147Sm | 1 |
| ± | 0.05 |
| 238Ft | 0.7 |
| ± | 0.05 |
| 235Ft | 0.7 |
| ± | 0.05 |
| 232Ft | 0.7 |
| ± | 0.05 |
| 147Ft | 0.7 |
| ± | 0.05 |





| Column header | Example optional data input |
|---|---|
| r 238U-232Th | 0.1 |
| r 238U-147Sm | 0.1 |
| r 232Th-147Sm | 0.1 |
| r 238Ft-235Ft | 0.9 |
| r 238Ft-232Ft | 0.9 |
| r 238Ft-147Ft | 0.9 |
| r 235Ft-232Ft | 0.9 |
| r 235Ft-147Ft | 0.9 |
| r 232Ft-147Ft | 0.9 |

**Table 1: Example input column names. All values in this table are purely for illustration and do not reflect actual data. The values for uncertainty covariance are given by their Pearson correlation coefficient ($r$; Eq. (23)). If no values are provided, uncertainties are assumed to be independent.**

**4.2 Output**

There are two main outputs from HeCalc: the results of the date calculation and uncertainty propagation, and the
histograms of the Monte Carlo results for each sample. At a minimum, the sample name, raw date, and corrected date are saved
to an Excel sheet titled "Uncertainty Output" that includes a header with the input file's name and directory. The raw and
corrected date in these columns is calculated using each exact input value (e.g., mol 238U = 1 in Table 1); we refer to these
dates as "nominal dates" below. The selection of linear uncertainty propagation causes columns to be added titled "Linear raw
uncertainty" and "Linear corrected uncertainty" for the linear error propagation results without and with alpha ejection
correction, respectively. If Monte Carlo error propagation is selected, a header line specifying the user-requested precision is
added, and the columns "MC average CI, raw", "MC +68% CI, raw", "MC -68% CI, raw", and the corresponding values for
$F_T$-corrected dates (titled with "corrected" instead of "raw") are included along with a column giving the number of Monte
Carlo simulations run. The confidence intervals are reported as the 15.865 and 84.135 percentiles (the 68% confidence interval)
of the Monte Carlo results, converted to uncertainty values by reference to the nominal date. Throughout this manuscript, the
asymmetry of the confidence intervals will be calculated with respect to the nominal date calculation. It is worth noting that
the nominal date does not strictly correspond to the mode of the histogram, and instead falls toward the skewed side, meaning
that the skew calculations presented here are a slight underestimate of the actual asymmetry in the distribution.

If the user chooses to include histograms in the output, an Excel sheet titled "Histogram Output" is added to the
workbook, with columns for the center of each histogram bin (i.e., the individual intervals in the histogram) and number of
simulations in that bin as x- and y- values for the both the raw and $F_T$-corrected dates. Four total columns are therefore present
for each sample. The number of bins is equal to 1/1000[th] the number of simulations run or ten bins, whichever is greater. If





parameterization is selected, the histogram is fit to a skew-normal distribution. Although this distribution does not perfectly replicate the histograms generated by HeCalc, it allows for first-order interpretations using continuous probability distributions. Columns are appended to the end of the "Uncertainty Output" sheet titled "Hist raw fit a", "Hist raw fit u", "Hist raw fit s",

and the corresponding values for $F_T$-corrected calculations. These parameters correspond to the shape ("a", the skewness), location ("u", a measure of central tendency), and scale ("s", the width of the distribution) parameters for a skew-normal distribution probability distribution function (Azzalini, 1985; O'Hagan and Leonard, 1976).

| Output Header | Example output: | Included when: |
|---|---|---|
| Sample | Sample1 | Always |
| Raw date | 62.4 | Always |
| Linear raw uncertainty | 2.65 | Linear propagation selected |
| MC average CI, raw | 2.66 | Monte Carlo propagation selected |
| MC +68% CI, raw | 2.77 | Monte Carlo propagation selected |
| MC -68% CI, raw | 2.55 | Monte Carlo propagation selected |
| Corrected date | 88.96 | Always |
| Linear corrected uncertainty | 6.31 | Linear propagation selected |
| MC average CI, corrected | 6.34 | Monte Carlo propagation selected |
| MC +68% CI, corrected | 6.78 | Monte Carlo propagation selected |
| MC -68% CI, corrected | 5.9 | Monte Carlo propagation selected |
| Number of Monte Carlo simulations | 502553 | Monte Carlo propagation selected |
| Hist raw fit a | 1.29 | Parameterization selected (requires Monte Carlo) |



| | | |
|---|---|---|
| Hist raw fit u | 60.34 | Parameterization selected (requires Monte Carlo) |
| Hist raw fit s | 3.44 | Parameterization selected (requires Monte Carlo) |
| Hist corrected fit a | 1.51 | Parameterization selected (requires Monte Carlo) |
| Hist corrected fit u | 83.72 | Parameterization selected (requires Monte Carlo) |
| Hist corrected fit s | 8.55 | Parameterization selected (requires Monte Carlo) |

**Table 2: HeCalc output column headers, produced by Table 1 inputs. The header for the file will contain a line for the file path of the input file and (if Monte Carlo propagation is selected) the user-requested precision.**

## 5 Uncertainty behavior

Below, we first carry out an analysis to explore how analytical and geometric input uncertainties influence the overall behavior of date uncertainty, skew in the Monte Carlo results, and differences between date uncertainties derived from the Monte Carlo and linear uncertainty propagation methods as a function of the measured date. We then use a compilation of real data to examine these trends in uncertainty for typical (U-Th)/He data.

### 5.1 Uncertainty in date as a function of input uncertainties

We examined the overall behavior of date uncertainty from 0 to 4.6 Ga as a function of relative input uncertainties of 1%, 5% and 20% on $^4$He (Fig. 3a), radionuclides (Fig. 3b) and isotope-specific $F_T$ values (Fig. 3c). This range of dates was generated by fixing the $^{238}$U and $^{232}$Th values while varying $^4$He values (no $^{147}$Sm was included because of its generally negligible influence on apatite and zircon results). Th/U ratios representative of a typical apatite (from a compilation of apatite data; Sect. 5.4), a typical zircon (based on the Fish Canyon Tuff zircon reference standard), and the Durango apatite reference standard (0.6, 1.25, and 16.1, respectively) were used. For all calculations, an isotope-specific $F_T$ value of 0.7 was applied to all isotopes to permit comparisons between raw and $F_T$-corrected dates (while isotope-specific values will differ in real data, we simplify these to a single value here). We initially explored the influence of individual uncertainties on the date by varying the relative uncertainty of one input parameter ($^4$He, radionuclides, or $F_T$) while fixing all other uncertainties at 0 (Fig. 3). We then evaluated how combinations of input uncertainties can influence the date (Fig. 4), although this is more fully evaluated in practical terms using real data, as in Sect. 5.4.

For these exercises, we use the results from Monte Carlo uncertainty propagation, as this technique is in theory fully accurate (see Sect. 5.3 for further discussion). We used a constant number of simulations set at $10^8$ to provide precise estimates





of skew and comparisons between the Monte Carlo and linear uncertainty propagation methods. This number of simulations
corresponds to a minimum precision of the mean date of ~0.0002% (2 ppm).

For individual input uncertainties, at young dates the input and output relative uncertainties are similar. If all
uncertainty is in the helium value or correlated $F_T$ values, the relative date uncertainty is equivalent to the input uncertainty at
zero age (Fig. 3a&c). For uncertainty in the radionuclides, and for uncorrelated $F_T$ values, the relative date uncertainty at zero
age is approximately 80% the magnitude of the relative input uncertainty (a 4:5 ratio). The exact scaling between input and
output uncertainties is dependent on the Th/U ratio (Fig. 3b&c). Date uncertainty associated with uncorrelated $F_T$ uncertainty
behaves empirically in much the same way as uncertainty from radionuclide measurements, which is to be expected given that
$F_T$ and radionuclide values are mathematically equivalent in the (U-Th)/He date equation. The absolute amount of $^4$He and/or
radionuclides is unimportant; the results are identical for a given date (i.e., a given $^4$He/radionuclide ratio), indicating that for
very young samples with low $^4$He, the uncertainty budget in $^4$He may dominate the date uncertainty. In addition, corrected and
raw date uncertainties are identical; for the same input uncertainties (excluding uncertainty in $F_T$), the same date uncertainty
is observed after $F_T$ correction is applied.







**Figure 3: Corrected (U-Th)/He date uncertainties for dates of 0 to 4.6 Ga with input uncertainty on only one parameter and all others held at zero.** Plots of relative uncertainty (in percent) of the corrected (U-Th)/He date calculated by Monte Carlo uncertainty propagation vs. corrected (U-Th)/He date for A) uncertainty only in $^4$He, B) uncertainties only in radionuclides, and C) uncertainties only in $F_T$, where fully correlated uncertainties ($r = 1$) are shown with solid lines and uncorrelated uncertainties ($r = 0$) are shown with dashed lines. Relative input uncertainties of 1% (top panels), 5% (middle panels), and 20% (bottom panels) were applied. The two 68% confidence intervals of the distributions resulting from Monte Carlo simulation were averaged to derive an equivalent 1σ uncertainty. The line colors correspond to the Th/U ratio for typical apatite (0.61, orange curve; derived from apatite data compilation), for typical zircon (1.25, green curve, derived from the Fish Canyon Tuff zircon reference standard), and for the Durango apatite reference standard (16.1, blue curve).

For all input uncertainties, the relative date uncertainty decreases with increasing absolute date. While uncertainty in $^4$He has a one-to-one relationship with date uncertainty at zero age, at 4.6 Ga the date uncertainty is approximately half that of





the input $^4$He uncertainty (Fig. 3a). The same phenomenon is observed for uncertainty in radionuclides and $F_T$ (Fig. 3b&c).

The relative extent of decreasing uncertainty as a function of increasing date is dependent on Th/U ratio and is independent of the magnitude of input uncertainty (i.e., the three vertically stacked panels in Fig. 3a, 3b, and 3c are identical aside from the scale of their y-axes).

Similar trends are observed when uncertainty is included in multiple input parameters (Fig. 4). At zero age, the uncertainty on the date introduced by each input parameter combines roughly in quadrature to provide the uncertainty on the

date, subject to the 80%, or 4:5 ratio, output uncertainty for the radionuclides (note that only correlated $F_T$ uncertainties are included in Fig. 4). For example, with 5% input uncertainty in $^4$He (which alone introduces 5% uncertainty in the date at zero age) and 5% uncertainty on the radionuclides (which alone introduces ~4% uncertainty in the date), the output date uncertainty combines these in quadrature to give an output uncertainty of 6.4% ($\sqrt{0.05^2 + 0.04^2} \cong 0.064$; dashed line, Fig. 4b). Likewise, at zero age, a 5% uncertainty in all parameters ($^4$He, radionuclides, correlated $F_T$ uncertainty), each of which alone introduces

a date uncertainty of 5%, 4%, and 5%, respectively, together yield a date uncertainty of 8.1% ($\sqrt{0.05^2 + 0.04^2 + 0.05^2} \cong 0.081$; solid line, Fig. 4b). Decreasing uncertainty with increasing date is also observed for multiple input uncertainties, with individual uncertainties combining in the manner described above. That is, a 5% uncertainty in $^4$He and in the radionuclides at 4.6 Ga each individually result in date uncertainties of ~2%, which when combined in quadrature yield a combined date uncertainty of 2.8% ($\sqrt{0.02^2 + 0.02^2} \cong 0.028$; far right portion of dashed line in Fig. 4b).



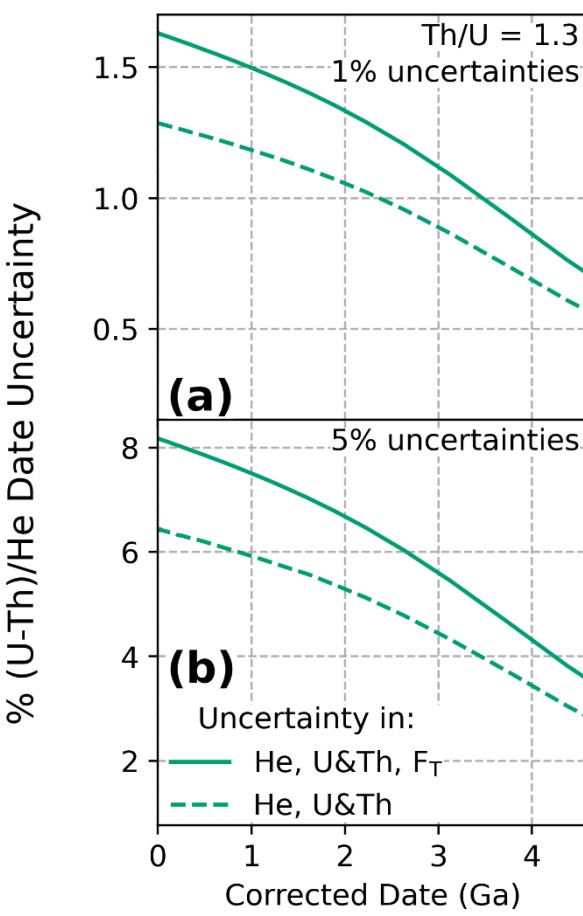


**Figure 4: Uncertainty in the corrected date resulting from uncertainty in multiple input parameters. This figure shows the combination of A) 1% uncertainty and B) 5% uncertainty in $^4$He and radionuclides, as well as all input values including fully correlated uncertainty in $F_T$ ($r = 1$) for a Th/U ratio of 1.3 (the green curve, derived from the Fish Canyon Tuff zircon reference standard). The solid line includes uncertainty in all parameters, and dashed line includes uncertainty in $^4$He and radionuclides.**

Uncertainties that are combined in quadrature may have unexpected properties for some practitioners. For instance, when combining uncertainties with equal magnitude, the resulting uncertainty will be only ~1.4 times larger than the input, rather than twice as large as might be expected. Alternatively, if input uncertainties have highly differing magnitudes, the larger uncertainty will dominate and the resulting combined uncertainty will be approximately equal to the larger uncertainty. As an example, a 10% and 1% uncertainty combined in quadrature will result a 10.05% uncertainty. This behavior suggests

that reducing the magnitude of the largest input uncertainty will be the most effective means of reducing overall date uncertainty.

The phenomenon of decreasing uncertainty with increasing date is a result of the "roll over" of the helium ingrowth curve due to its nature as an exponential function. Because of this roll over, constant uncertainty in the independent variable (i.e., $^4$He or the radionuclides) will correspond to smaller uncertainty in the dependent variable (the date) as the value of the





dependent variable increases. Fig. 5 is a schematic showing log plots of date vs. $^4$He and date versus $^{238}$U that illustrates this

phenomenon. For young dates, the exponential term in the date equation (Eq. (2)) approaches zero, meaning that the relative

uncertainties input to the He age equation will be roughly reflected in the output uncertainties (Sample 1, Fig. 5). For older

dates, this exponential term becomes increasingly large, resulting in roll over of the ingrowth curve and reducing the date

uncertainty relative to the inputs (Sample 2, Fig. 5). The exact form of this roll over is dictated by the relative abundance of

each radioisotope, resulting in the variations observed in Fig. 3 for differing Th/U values.

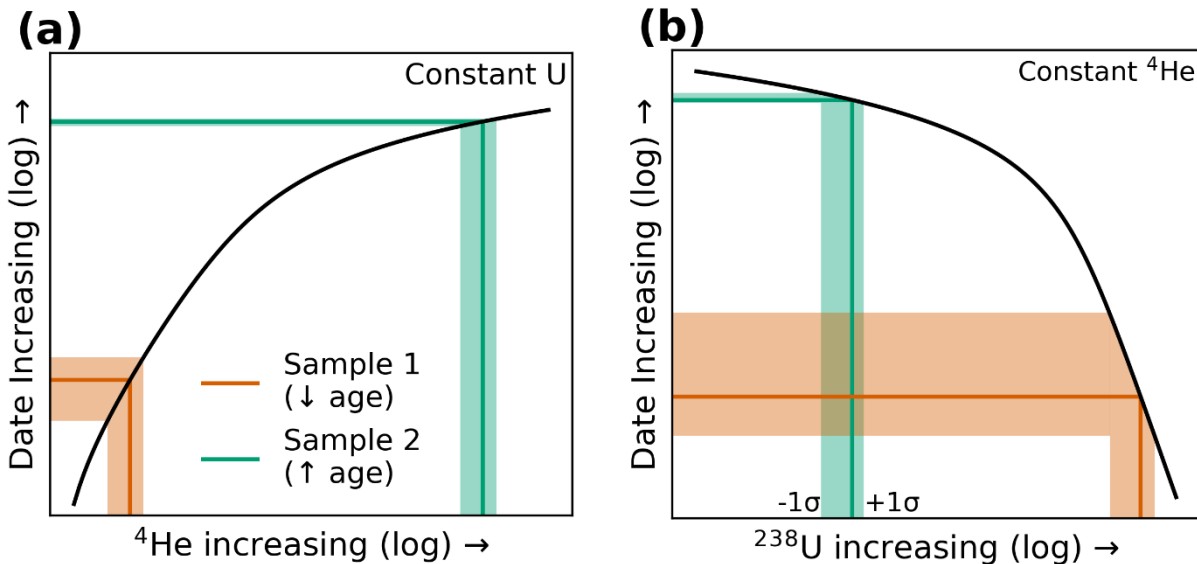

**Figure 5: A schematic showing how non-linearity in the (U-Th)/He date equation causes decreasing uncertainty with increasing date. Log-log plots are shown in black of A) Date increasing as a function of increasing $^4$He with other parameters fixed, and B) Date decreasing as a function of increasing $^{238}$U with other parameters fixed. Two example samples, one young (Sample 1, orange line) and one old (Sample 2, green line) are provided with gaussian constant relative uncertainty (1σ depicted by the shaded region). The apparent asymmetry in the uncertainty along the x-axis is a result of the logarithmic plot. The non-linearity of the (U-Th)/He age equation is exaggerated for this schematic by decreasing the uranium decay constant to improve visibility of its effects. Note that in log-log space, the spread (i.e., uncertainty) in the x-axis is constant for constant input uncertainty, but the resulting uncertainty on the y-axis shrinks with increasing date.**

## 5.2 Skewed distributions

Skew refers to the extent of asymmetry in the "tails" of a distribution (Fig. 6). For example, the skewed distribution

in Fig. 6c is highly asymmetrical, while the less skewed distribution in Fig. 6a is more symmetrical. This asymmetry would

most accurately be reported as separate positive and negative uncertainty values referring to the 68% confidence interval rather

than the more typical 1σ uncertainty reporting of symmetrical uncertainty (e.g., 100 [+11, -9] Ma instead of 100 ±10 Ma).

Although "skewness", *sensu stricto,* is a statistical concept referring to the third standardized moment of a population, this

metric is unitless and generally unintuitive, so here we report skew in HeCalc-generated histograms by taking the percent

difference between the positive and negative 68% confidence intervals with respect to the nominal date.





**Figure 6: An illustration of how differing uncertainty affects the skew of date probability distributions for inputs yielding a date of 15.1 Ma (assuming a typical apatite Th/U ratio of 0.61). A) Low radionuclide uncertainty of 1%, giving ~1.6% skew; B) high radionuclide uncertainty of 5%, giving ~8.5% skew; and C) extremely large radionuclide uncertainty of 20% (see discussion of real data in Sect. 5.4), giving ~38% skew. The gaussian fit in panels A and B are almost entirely concealed by the skew-normal fit plotted above it. The left column shows all distributions at the same scale, while the right-hand column zooms into the more precise (and less skewed) distributions to show detail.**





The magnitude of skew correlates directly with the magnitude of input uncertainty (Fig. 7). For low relative input uncertainties on all parameters, the magnitude of skew is low. For example, uncertainties of 1% for all inputs yield ≤2% skew for dates from 0 to 4.6 Ga (Fig. 7a-c, top panels). Only when the input uncertainties are larger does the effect of skew on the dates become substantial (Fig. 7a-c, middle and bottom panels). In the case of larger uncertainty in He (Fig. 7a, middle and bottom panels), skew increases from zero to progressively larger negative values at older dates. The inverse is true for

uncertainty in the radionuclides and $F_T$; skew is highest when uncertainty in these parameters is high for young dates and decreases with increasing age (Fig. 7b&c, middle and bottom panels). Note that although asymmetrical uncertainties as high as ~40% can be yielded by radionuclide uncertainties of 20%, such large uncertainties are anomalous and do not typify most high-quality (U-Th)/He datasets (Sect. 5.4).





**Figure 7: Illustration of the impact on skew from 0 to 4.6 Ga of varying individual relative input uncertainties while holding other uncertainties fixed at zero. Skew is shown as a percent difference between the 68% confidence intervals with respect to the nominal date value, as a function of input uncertainty for A) $^4$He, B) radionuclides, and C) $F_T$. Relative input uncertainties of 1% (top panels), 5% (middle panels), and 20% (bottom panels) were applied. The line colors correspond to the Th/U ratio for typical apatite (0.61, orange curve; derived from apatite data compilation), for typical zircon (1.25, green curve, derived from the Fish Canyon Tuff zircon reference standard), and for the Durango apatite reference standard (16.1, blue curve). Note that unlike Fig. 3, the y-axis scale is different for each panel.**

When uncertainty is included in multiple input parameters, the overall skew is a combination of the skew resulting from individual input uncertainties (Fig. 8). Unlike date uncertainty, which combines individual inputs in quadrature, the combination of skew from individual inputs does not follow an easily predictable trend. For input uncertainties of 1% and 5%





for [4]He and the radionuclides only, the skew is largest at zero age (~1% and 5%, respectively), and declines with increase age (dashed lines in Fig. 8). Because uncertainty in [4]He generates negative skew at older dates (Fig. 7a), the skew from these combined uncertainties becomes negative at dates ⪦3 Ga as the skew resulting from [4]He uncertainty overwhelms the skew from radionuclides, which has the opposite sign and is greatest at young dates (Fig. 7b). Similarly, for input uncertainties of 1% and 5% for all parameters (including $F_T$), the skew is largest at zero age (~2% and ~11% respectively) and declines with

decreasing age (to ~0% at 4.6 Ga; solid lines in Fig. 8).

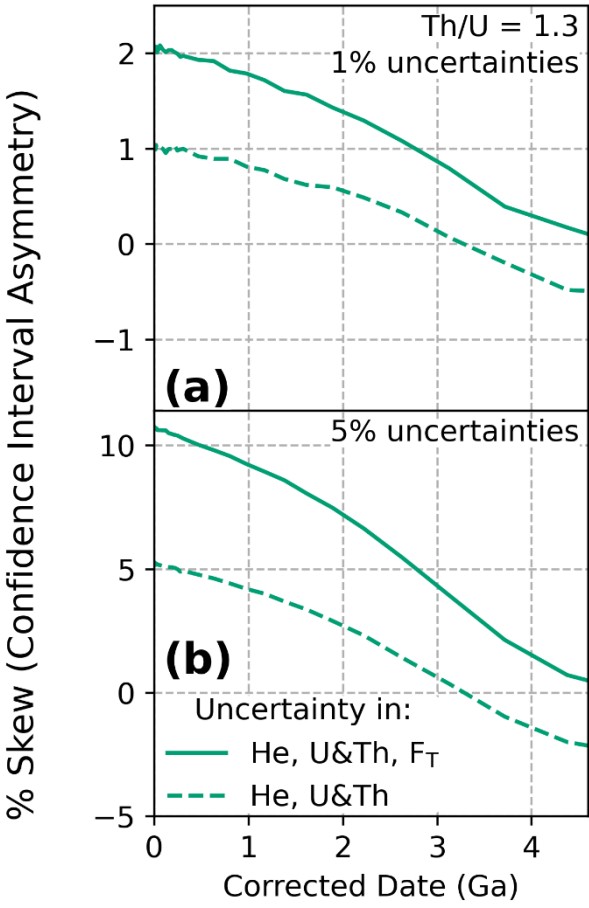

**Figure 8: The skew in the date probability distribution resulting from combining multiple input uncertainties. Skew is shown as a percent difference between the 68% confidence intervals. This figure shows each possible combination of A) 1% uncertainty and B) 5% uncertainty in [4]He, radionuclides, and fully correlated $F_T$ ($r = 1$) for a Th/U ratio of 1.3 (the green curve, derived from the Fish**

**Canyon Tuff zircon reference standard). The solid line includes uncertainty in all parameters, and the dashed line includes uncertainty in just [4]He and the radionuclides.**

        Much like the decrease in date uncertainty with increasing date for constant input uncertainties, skew occurs as a result of the "roll over" of the He age equation due to its exponential nature. For large absolute uncertainty values, a significant portion of the age curve is captured within the uncertainty, increasing the amount of non-linearity contained within this



uncertainty, resulting in skew. Given constant relative input uncertainty, the absolute uncertainty will therefore be largest at the largest input values, corresponding to younger dates for the radionuclides and $F_T$ values, and more ancient dates for $^4$He values.

Uncertainty in He and radionuclides therefore produce opposing skew effects because they have an inverse relationship with respect to the age curve: an increase in He results in an increasing date, while decreasing radionuclide
concentration results in an increasing date (Fig. 5). This relationship causes the increasing and negative skew with age for large helium uncertainties (Fig. 7a) and the positive and decreasing skew with increasing age for large eU uncertainties (Fig. 7b). That is, skew is largest for both $^4$He and radionuclides when the absolute value and uncertainty are largest—at older dates for $^4$He and younger dates for the radionuclides.

**5.3 Comparison of Monte Carlo and linear uncertainty propagation**

To compare linear and Monte Carlo error propagation derived uncertainties, we average the two 68% confidence intervals to determine uncertainty from both methods at the 1σ level. For data with high skew, this method provides a means of comparing the scale of these two differing output distributions directly. The magnitude of the error in uncertainty estimation from linear uncertainty propagation due to nonlinearity in the date equation is proportional to the magnitude of the input uncertainties. As shown in Fig. 9a, for uncertainty in He alone, the Monte Carlo and linear methods yield identical results at
younger dates, with linear uncertainty propagation beginning to underestimate the true uncertainty values at older dates as the absolute magnitude of $^4$He uncertainty increases (reaching a maximum discrepancy of ~2% for input uncertainties of 20% at 4.6 Ga). Uncertainty in radionuclides and $F_T$ have the opposite effect; the discrepancy between the Monte Carlo and linear methods is greatest (~3% for input uncertainties of 20%, dependent on Th/U ratio and correlation in $F_T$ uncertainties) at zero age and decreases with increasing date. This small extent of error indicates that Monte Carlo and linear methods are in general
agreement.





**Figure 9: The extent of error introduced by the use of linear uncertainty propagation instead of Monte Carlo uncertainty propagation for dates from 0 to 4.6 Ga. Error is shown for uncertainty in A) $^4$He, B) radionuclides, and C) $F_T$ using the percent difference between the Monte Carlo and linear uncertainty propagation results, with the average 68% confidence intervals used to represent a single uncertainty value for the Monte Carlo results. Relative input uncertainties of 1% (top panels), 5% (middle panels), and 20% (bottom panels) were applied. The line colors correspond to the Th/U ratio for typical apatite (0.61, orange curve; derived from apatite data compilation), for typical zircon (1.25, green curve, derived from the Fish Canyon Tuff zircon reference standard), and for the Durango apatite reference standard (16.1, blue curve). Note that unlike Fig. 3, the y-axis scale is different for each panel.**

As linear uncertainty propagation relies on an arithmetic calculation rather than random sampling, this method provides predictable and repeatable results for uncertainty calculations and is amenable to encoding in spreadsheet programs, facilitating the inclusion of the equations provided in Sect. 3.2 in existing spreadsheet-based workflows. However, the presence



of skew in (U-Th)/He date uncertainties and the inaccuracies in uncertainty calculation induced by non-linearity in the (U-Th)/He age equation indicate that the more accurate Monte Carlo uncertainty propagation method is more universally applicable. Although in the past computational (in)efficiency was generally considered the weakness of Monte Carlo methods, running as many as one million Monte Carlo simulations in HeCalc takes less than one second on a modern computer for a typical sample. This number of random samples provides a sufficiently large population that output histograms are relatively smooth, and results in accurate calculations of uncertainty with sufficient significant figures that the model-to-model variation induced by random sampling is negligible. As the Monte Carlo method in HeCalc is not excessively computationally intensive and provides both skew and accurate uncertainty calculations, we suggest that the Monte Carlo method is preferable to linear uncertainty propagation in the (U-Th)/He system.

### 5.4 Uncertainty in real data

### 5.4.1 Uncertainty budget in real data

In the preceding sections, we explored the impacts of theoretical input uncertainties on the overall uncertainty budget in the (U-Th)/He system, evaluated the influence on skew, and compared the two methods of uncertainty propagation discussed. However, most pertinent to day-to-day (U-Th)/He analyses are how typical $^4$He, radionuclide, and $F_T$ uncertainties impact date uncertainties and interpretation of (U-Th)/He data. To assess the typical values for each of the uncertainty components described in Sect. 2 for the most commonly analyzed minerals, we assembled a compilation of apatite and zircon data acquired using common (U-Th)/He methods and instrumentation (quadrupole noble gas mass spectrometer and quadrupole ICP-MS). For consistency, all data included this compilation were analyzed using identical instrumentation and methods in the University of Colorado Thermochronology Research and Instrumentation Laboratory (CU TRaIL). These data include 1,978 apatite analyses following the methods of Sturrock et al. (2021) and 1,753 zircon analyses using the methods described in Peak et al. (2021).

These data are depicted in Fig. 10, showing the distribution of percent relative uncertainty in $^4$He, $^{238}$U, $^{232}$Th, and (for apatite) $^{147}$Sm. To best represent these distributions, we take the median value and 68% confidence interval calculated using the percentile approach described in Sect. 3.3, which are shown in Table 3. These results indicate that the uncertainty for each measured value is lower in a typical zircon analysis than in a typical apatite analysis, likely due to the higher concentration of radionuclides in a typical zircon grain and the greater retentivity of $^4$He, resulting in analytical measurements that have greater count rates and are less impacted by the uncertainty associated with blank and background levels.





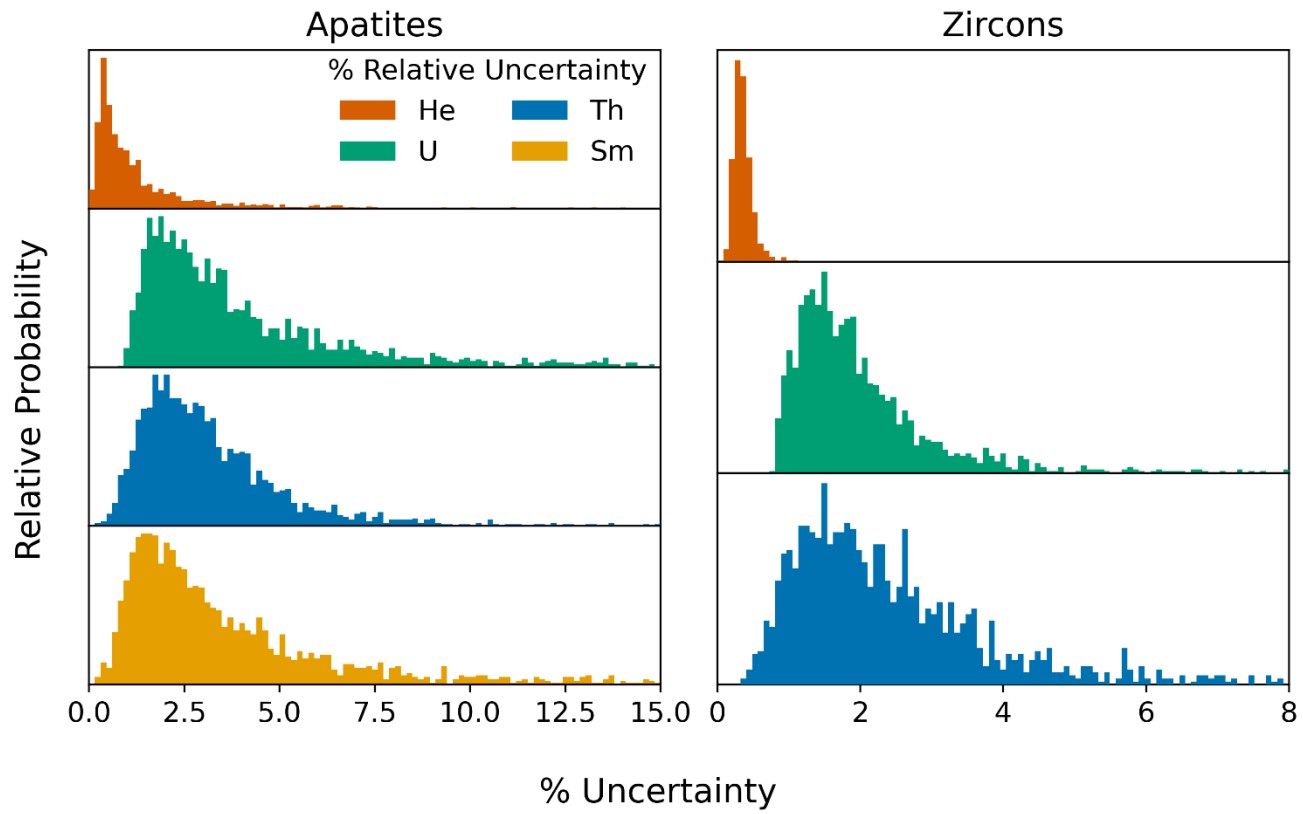

**Figure 10: Histograms of uncertainty in all $^4$He and radionuclide data collected in CU TRaIL from October 2017 to March 2020, depicted as a percent uncertainty relative to each absolute datum.**

| Uncertainty component | Apatite (% uncertainty) | Zircon (% uncertainty) |
|---|---|---|
| $^4$He | 0.86 [+1.9, -0.51] | 0.34 [+0.14, -0.10] |
| $^{238}$U | 3.2 [+3.6, -1.4] | 1.8 [+1.1, -0.6] |
| $^{232}$Th | 2.8 [+2.0, -1.2] | 2.2 [+2.3, -1.0] |
| $^{147}$Sm | 2.8 [+4.4, -1.4] | N.M. |

**Table 3: Median and 68% percentile confidence interval (15.865 and 84.135 percentile) values for data from CU TRaIL. N.M. = "Not Measured"**

In this dataset, uncertainty in the quantification of radionuclides dominates relative to uncertainty in $^4$He measurement. The uncertainty in apatite $^{238}$U analyses is 3.2 [+3.6, -1.4]% (shown as median [+68% Confidence interval, -68% confidence interval]), for $^{232}$Th is 2.8 [+2.0, -1.2]%, and for $^{147}$Sm 2.8 [+4.4, -1.4]%. In comparison, the $^4$He data is approximately 3 times more precise, with a relative uncertainty of 0.86 [+1.9, -0.51]%. The same pattern of radionuclide uncertainty greater than He uncertainties also holds for the zircon data. For zircons, the radionuclide measurements are about




half again more precise than for apatite, with $^{238}U$ and $^{232}Th$ values of 1.8 [+1.1, -0.6]% and 2.2 [+2.3, -1.0]%. Similarly, the
$^4$He uncertainty of 0.34 [+0.14, -0.10]% for zircon is ~3 times more precise than for apatite.

Using HeCalc, we analyzed the uncertainty in these data using both linear and Monte Carlo approaches to determine their distribution of date uncertainty (Fig. 11). For apatite, propagating only uncertainties on $^4$He and radionuclides (i.e., "analytical" uncertainties) yields date uncertainties of 2.9 [+3.1, -1.2]%. Zircon dates are generally more precise with uncertainties of 1.7 [+1.1, -0.5]% (Fig. 11b, Table 4).

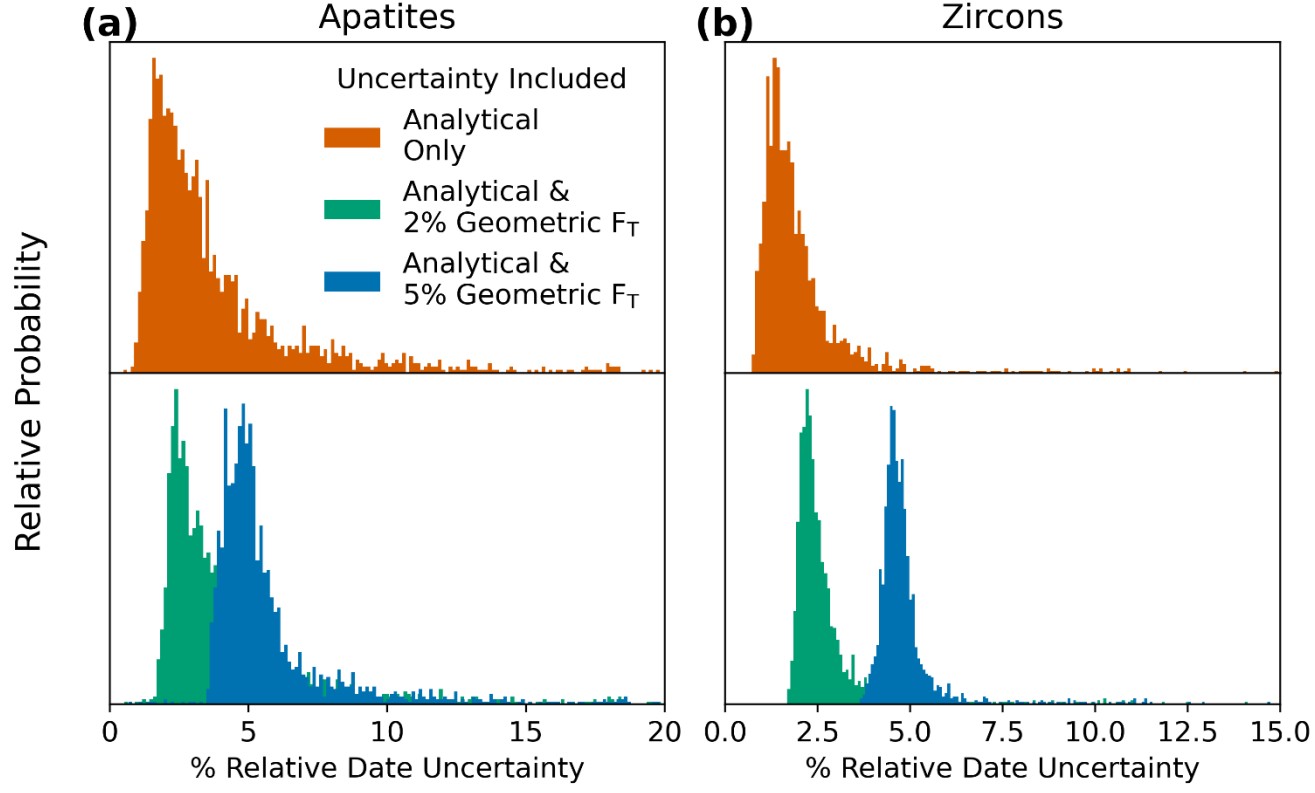


**Figure 11: Histograms of percent relative uncertainty for apatite and zircon analyses in the CU TRaIL from 2017-2020. The top panels show input uncertainties for only analytical uncertainties ($^4$He and radionuclides), while the lower panels additionally include 2% (green) or 5% (blue) geometric uncertainty in F$_T$, assuming fully correlated individual F$_T$ uncertainty values ($r = 1$).**

| Included Uncertainty Components | Apatite Date Uncertainty (%) | Zircon Date Uncertainty (%) |
|---|---|---|
| Analytical Only | 2.9 [+3.1, -1.2] | 1.7 [+1.1, -0.5] |
| Analytical & 2% Geometric | 3.3 [+2.9, -1.0] | 2.9 [+0.9, -0.3] |
| Analytical & 5% Geometric | 5.0 [+2.2, -0.8] | 4.7 [+0.5, -0.4] |

**Table 4: Median and 68% percentile confidence interval (15.865 and 84.135 percentile) percent date uncertainty for reduced data**
**from CU TRaIL. Analytical uncertainty refers to uncertainty in $^4$He and radionuclides.**



As discussed in Sect. 2, the uncertainties associated with $F_T$ values are not currently well constrained, but estimates of uncertainty stemming from geometry alone are ~2-9% (Cooperdock et al., 2019; Evans et al., 2008). These uncertainties are also likely highly correlated (for these analyses we make the simplifying assumption of perfect correlation, $r = 1$). Notably, these inferred $F_T$ uncertainties are within the same order of magnitude as the analytical uncertainties presented in Table 3. We

included 2% and 5% uncertainty in $F_T$ in a second analysis of this data compilation to obtain initial estimates of the influence of $F_T$ uncertainty on date uncertainty (green and blue in Fig. 11; Table 4).

With uncertainty in $F_T$ included, the date uncertainty increases substantially. For apatites, the uncertainty value increases from 2.9 [+3.1, -1.2]% for analytical ($^4$He and radionuclide) uncertainties alone to 3.3 [+2.9, -1.0]% and 5.0 [+2.2, -0.8]%, respectively, when also including $F_T$ uncertainties of 2% or 5%. For zircons, the uncertainty increases from 1.7 [+1.1,

-0.5]% to 2.4 [+0.9, -0.3]% and 4.7 [+0.5, -0.4]%, respectively. The addition of constant 2% uncertainty in $F_T$ values most heavily impacts the more precise analytical measurements because $F_T$ uncertainty comprises a correspondingly large proportion of the uncertainty budget. Similarly, the inclusion of 5% uncertainty in $F_T$ overwhelms most other uncertainty components, resulting in date uncertainties near 5%. Given that initial estimates of geometric uncertainty in $F_T$ are on the same order of magnitude as—and potentially larger than—typical analytical uncertainties in (U-Th)/He dating, further effort to

constrain $F_T$ uncertainty across a wide range of characteristics (i.e., on minerals other than apatite such as zircon and titanite, and a complete range of grain shapes) is important to fully capture the anticipated intra-sample variability in (U-Th)/He data.

### 5.4.2 Skew and error in linear uncertainty propagation for real data

Using HeCalc we also analyzed the compilation of real data in Fig. 10 for skew and deviation of the linear uncertainty propagation outputs from those using the Monte Carlo method. Positive skew is common in the compiled dataset and can be

significant, as would be predicted based on the patterns in skew for theoretical data (Fig. 7) and the real data uncertainties in Table 3. With only analytical uncertainty included, skew in apatites is 4.4 [+4.3, -2.0]% and in zircons is 3.2 [+2.3, -1.0]%. Much like date uncertainties, the inclusion of uncertainty in the $F_T$ parameter causes an increase in skew (Fig. 12a&b, Table 5). For apatites, skew rises to 5.5 [+3.9, -1.6]% and 9.7 [+2.5, -1.0]% for 2% and 5% uncertainty in $F_T$, respectively. For zircons, the same combinations of uncertainty yield skew of 4.8 [+1.7, -0.7]% and 9.7 [+1.0, -0.8]%. With 2% $F_T$ uncertainty,

approximately 14% of apatite data have an asymmetrical uncertainty of 10% or greater. For zircon, ~5% of all data with 2% $F_T$ uncertainty included have a skew of 10% or greater. As an example with 10% skew, a typical 100±6.4 Ma date should instead be presented as 100 [+6.7, -6.1] Ma when uncertainty is propagated to include asymmetrical uncertainties.




**Figure 12: Histograms of skew as percent asymmetry in the 68% confidence intervals, and the % difference between averaged Monte**
**Carlo-derived confidence intervals and linear uncertainty propagation. Input uncertainties include analytical uncertainty only ($^4$He**
**and radionuclides, orange), and analytical uncertainties propagated with 2% (green) or 5% (blue) geometric uncertainty in $F_T$,**
**assuming fully correlated individual $F_T$ uncertainty values ($r = 1$). The outlines of covered histograms are included to show detail**
**for each.**

General practice in (U-Th)/He dating has been to report symmetrical uncertainties; for many cases, averaging of

asymmetrical uncertainties in data reporting is unlikely to significantly impact interpretations. However, for highly

asymmetrical uncertainties, it may be appropriate to report positive and negative uncertainties separately, and only combine

the reported uncertainties if they are indistinguishable within the appropriate number of significant figures. Our results suggest

that skew may be an important consideration when interpreting some (U-Th)/He data, particularly less precise data, as the

larger asymmetries in uncertainty discussed here may be important to determinations of whether a set of data is consistent with

a given hypothesis within uncertainty. However, a challenge to interpreting data with asymmetrical uncertainties is that no

widely used inverse thermal history modeling software for (U-Th)/He data permits the input of asymmetrical uncertainty.



Future work implementing skewed probability distributions in such software may enhance interpretation of the subset of (U-Th)/He data characterized by highly skewed uncertainties.

Error due to linear uncertainty propagation (i.e., inaccurately calculated uncertainty resulting from an assumption of linearity) is present but largely insignificant in this dataset. When errors in linear uncertainty propagation are quantified with respect to the averaged 68% confidence intervals from Monte Carlo propagation, nearly all analyses have Monte Carlo- and linear-derived uncertainties within 1% of each other, regardless of the amount of $F_T$ uncertainty included, although slightly greater error is observed with 5% $F_T$ uncertainty included (Fig. 12c&d and Table 4).

This analysis of real data suggests that Monte Carlo uncertainty propagation provides improved uncertainty calculations relative to linear uncertainty propagation, particularly as a means of constraining skew in (U-Th)/He date uncertainty. Skew is likely more important for accurate uncertainty analysis than error introduced in the uncertainty calculation as a result of using linear uncertainty propagation. For the most common uncertainties in (U-Th)/He dating, the date uncertainties generated by Monte Carlo and linear uncertainty propagation are likely to be interchangeable. However, for a subset of samples with atypically large input uncertainties, the skew revealed by Monte Carlo uncertainty propagation may be

important to consider for date interpretation. Evaluating the magnitude of skew is easily achieved by using HeCalc for uncertainty propagation, providing improved confidence in (U-Th)/He date uncertainty calculation and interpretation.

| Included Uncertainty Components | Apatite Skew (%) | Zircon Skew (%) |
|---|---|---|
| Analytical Only | 4.4 [+4.3, -2.0] | 3.2 [+2.3, -1.0] |
| Analytical & 2% Geometric | 5.5 [+3.9, -1.6] | 4.8 [+1.7, -0.7] |
| Analytical & 5% Geometric | 9.7 [+2.5, -1.9] | 9.7 [+1.0, -0.8] |

**Table 5: Median and 68% percentile confidence interval (15.865 and 84.135 percentile) percent skew for reduced data from CU TRaIL. Analytical uncertainty refers to uncertainty in $^4$He and radionuclides.**

| Included Uncertainty Components | Apatite Linear Propagation Error (%) | Zircon Linear Propagation Error (%) |
|---|---|---|
| Analytical Only | -0.084 [+0.11, -0.17] | -0.047 [+0.10, -0.11] |
| Analytical & 2% Geometric | -0.11 [+0.12, -0.16] | -0.075 [+0.10, -0.12] |
| Analytical & 5% Geometric | -0.23 [+0.07, -0.15] | -0.22 [+0.05, -0.06] |

**Table 6: Median and 68% percentile confidence interval (15.865 and 84.135 percentile) percent linear uncertainty propagation error for reduced data from CU TRaIL. Analytical uncertainty refers to uncertainty in $^4$He and radionuclides. Percent linear error is calculated by the difference between average 68% confidence interval for Monte Carlo and the 1σ linear uncertainty**

## 6 Conclusions

      Here we publish fully traceable end-to-end calculations of uncertainty in (U-Th)/He dates, including the propagation of uncertainties in $F_T$ values. We also provide a software package, HeCalc, to do these calculations explicitly and to perform





more accurate Monte Carlo propagation of these uncertainties. Using this software package to perform a sensitivity analysis

of the uncertainty components in (U-Th)/He dating, we find that relative uncertainties become smaller for older (U-Th)/He

dates. Skewed (asymmetrical) date probability distributions are also possible, particularly for less precise data. For uncertainty

in radionuclide and $F_T$ values, skew is positive and highest at young dates, while for uncertainty in $^4$He, skew is negative and

largest at ancient dates. A comparison between the Monte Carlo and linear uncertainty propagation methods indicates that,

correcting for skew, both methods yield nearly identical results, with minor errors in the linear uncertainty propagation method.

These effects (falling uncertainty with increasing date, skew, linear uncertainty propagation error) are a result of non-linearity

in the (U-Th)/He date equation, and in the case of skew and linear propagation error, the fact that the equation is non-linear at

the scale of uncertainties in this system.

        Using a compilation of (U-Th)/He apatite and zircon analyses, we find that for a common instrumental setup

(quadrupole noble gas and ICP mass spectrometers), uncertainty in radionuclide quantification is generally 3-5 times larger

than the uncertainty in $^4$He measurement. The resulting typical date uncertainty is 2.9 [+3.1, -1.2]% of the measured value for

apatites and 1.7 [+1.1, -0.5]% for zircons. The inclusion of preliminary 2% and 5% geometric uncertainty in the $F_T$ values (and

assuming that these uncertainty values are fully correlated) yields greater date uncertainty of 3.3 [+2.9, -1.0]% and 5.0 [+2.2,

-0.8]% for apatites and 2.4 [+0.9, -0.3]% and 4.7 [+0.5, -0.4]% for zircons. For these data, the asymmetry in the 68% confidence

interval can be significant. With 2% uncertainty included in $F_T$, 14% of all apatite and 5% of all zircon analyses have a skew

of greater than 10%. The results of linear uncertainty propagation for these data agrees with the results from Monte Carlo

uncertainty propagation to within 1%, indicating that this error is likely negligible for nearly all data.

        Given that Monte Carlo uncertainty propagation permits calculation of skewed probability distributions and does not

make an assumption of linearity in the (U-Th)/He age equation, we propose that this method should be preferred for uncertainty

calculation in (U-Th)/He data. However, the current lack of a means of including asymmetrical uncertainty in thermal

modeling, and the roughly equivalent symmetrical uncertainty values from Monte Carlo and linear uncertainty propagation

methods, indicates that the results are likely interchangeable for common workflows, pending advancements in the (U-Th)/He

method and interpretative models.

        The methods presented here allow for robust inter-laboratory data comparisons and retrospective data analyses by

providing a consistent means of quantifying the uncertainty budget of a given (U-Th)/He analysis. Further developments of

the (U-Th)/He technique are also facilitated by this study. In particular, this work suggests that continued refinement of $F_T$

uncertainty is warranted, and provides a framework into which those developments may be placed. Using the Monte Carlo

results, asymmetrical uncertainty may also be quantified, and could potentially be included in future versions of thermal

modeling software. Finally, fully accounting for analytical and geometric uncertainties will better isolate the magnitude of

overdispersion and promote more effective examination of its causes.



## Appendix A: Additional linear uncertainty propagation equations

Here we print the equations presented in Sect. 3.2 in their expanded forms, along with a set of equations that allows for direct quantification of $^{235}$U. First, the expanded form for each derivative is:

$$\frac{\partial f}{\partial\ ^4He} = \frac{1}{\left[\begin{array}{c} 8\lambda_{238}\ ^{238}F_T\ ^{238}Ue^{\lambda_{238}t_i} + \frac{7}{137.818}\lambda_{235}\ ^{235}F_T\ ^{238}Ue^{\lambda_{235}t_i} + \\ 6\lambda_{232}\ ^{232}F_T\ ^{232}The^{\lambda_{232}t_i} + \lambda_{147}\ ^{147}F_T\ ^{147}Sme^{\lambda_{147}t_i} \end{array}\right]}$$

(a1)

$$\frac{\partial f}{\partial\ ^{238}U} = -\frac{8\ ^{238}F_T\left(e^{\lambda_{238}t_i}-1\right) + \frac{7}{137.818}\ ^{235}F_T\left(e^{\lambda_{235}t_i}-1\right)}{\left[\begin{array}{c} 8\lambda_{238}\ ^{238}F_T\ ^{238}Ue^{\lambda_{238}t_i} + \frac{7}{137.818}\lambda_{235}\ ^{235}F_T\ ^{238}Ue^{\lambda_{235}t_i} + \\ 6\lambda_{232}\ ^{232}F_T\ ^{232}The^{\lambda_{232}t_i} + \lambda_{147}\ ^{147}F_T\ ^{147}Sme^{\lambda_{147}t_i} \end{array}\right]}$$

(a2)

$$\frac{\partial f}{\partial\ ^{232}Th} = -\frac{6\ ^{232}F_T\left(e^{\lambda_{232}t_i}-1\right)}{\left[\begin{array}{c} 8\lambda_{238}\ ^{238}F_T\ ^{238}Ue^{\lambda_{238}t_i} + \frac{7}{137.818}\lambda_{235}\ ^{235}F_T\ ^{238}Ue^{\lambda_{235}t_i} + \\ 6\lambda_{232}\ ^{232}F_T\ ^{232}The^{\lambda_{232}t_i} + \lambda_{147}\ ^{147}F_T\ ^{147}Sme^{\lambda_{147}t_i} \end{array}\right]}$$

(a3)


$$\frac{\partial f}{\partial\ ^{147}Sm} = -\frac{^{147}F_T\left(e^{\lambda_{147}t_i}-1\right)}{\left[\begin{array}{c} 8\lambda_{238}\ ^{238}F_T\ ^{238}Ue^{\lambda_{238}t_i} + \frac{7}{137.818}\lambda_{235}\ ^{235}F_T\ ^{238}Ue^{\lambda_{235}t_i} + \\ 6\lambda_{232}\ ^{232}F_T\ ^{232}The^{\lambda_{232}t_i} + \lambda_{147}\ ^{147}F_T\ ^{147}Sme^{\lambda_{147}t_i} \end{array}\right]}$$

(a4)

$$\frac{\partial f}{\partial\ ^{238}F_T} = -\frac{8\ ^{238}U\left(e^{\lambda_{238}t_i}-1\right)}{\left[\begin{array}{c} 8\lambda_{238}\ ^{238}F_T\ ^{238}Ue^{\lambda_{238}t_i} + \frac{7}{137.818}\lambda_{235}\ ^{235}F_T\ ^{238}Ue^{\lambda_{235}t_i} + \\ 6\lambda_{232}\ ^{232}F_T\ ^{232}The^{\lambda_{232}t_i} + \lambda_{147}\ ^{147}F_T\ ^{147}Sme^{\lambda_{147}t_i} \end{array}\right]}$$

(a5)

$$\frac{\partial f}{\partial\ ^{235}F_T} = -\frac{7\ ^{235}U\left(e^{\lambda_{235}t_i}-1\right)}{\left[\begin{array}{c} 8\lambda_{238}\ ^{238}F_T\ ^{238}Ue^{\lambda_{238}t_i} + \frac{7}{137.818}\lambda_{235}\ ^{235}F_T\ ^{238}Ue^{\lambda_{235}t_i} + \\ 6\lambda_{232}\ ^{232}F_T\ ^{232}The^{\lambda_{232}t_i} + \lambda_{147}\ ^{147}F_T\ ^{147}Sme^{\lambda_{147}t_i} \end{array}\right]}$$

(a6)





$$\frac{\partial f}{\partial^{232}F_T} = -\frac{6\,^{232}Th\left(e^{\lambda_{232}t_i}-1\right)}{\left[\begin{array}{c}8\lambda_{238}\,^{238}F_T\,^{238}Ue^{\lambda_{238}t_i} + \frac{7}{137.818}\lambda_{235}\,^{235}F_T\,^{238}Ue^{\lambda_{235}t_i} + \\ 6\lambda_{232}\,^{232}F_T\,^{232}The^{\lambda_{232}t_i} + \lambda_{147}\,^{147}F_T\,^{147}Sme^{\lambda_{147}t_i}\end{array}\right]}$$

(a7)

$$\frac{\partial f}{\partial^{147}F_T} = -\frac{^{147}Sm\left(e^{\lambda_{147}t_i}-1\right)}{\left[\begin{array}{c}8\lambda_{238}\,^{238}F_T\,^{238}Ue^{\lambda_{238}t_i} + \frac{7}{137.818}\lambda_{235}\,^{235}F_T\,^{238}Ue^{\lambda_{235}t_i} + \\ 6\lambda_{232}\,^{232}F_T\,^{232}The^{\lambda_{232}t_i} + \lambda_{147}\,^{147}F_T\,^{147}Sme^{\lambda_{147}t_i}\end{array}\right]}$$

(a8)

If $^{235}U$ was directly quantified, the derivatives for $^{238}U$ and $^{235}U$ are

$$\frac{\partial f}{\partial^{238}U} = -\frac{8\,^{238}F_T\left(e^{\lambda_{238}t_i}-1\right)}{\left[\begin{array}{c}8\lambda_{238}\,^{238}F_T\,^{238}Ue^{\lambda_{238}t_i} + 7\lambda_{235}\,^{235}F_T\,^{235}Ue^{\lambda_{235}t_i} + \\ 6\lambda_{232}\,^{232}F_T\,^{232}The^{\lambda_{232}t_i} + \lambda_{147}\,^{147}F_T\,^{147}Sme^{\lambda_{147}t_i}\end{array}\right]}$$

(a9)

$$\frac{\partial f}{\partial^{235}U} = -\frac{7\,^{235}F_T\left(e^{\lambda_{235}t_i}-1\right)}{\left[\begin{array}{c}8\lambda_{238}\,^{238}F_T\,^{238}Ue^{\lambda_{238}t_i} + 7\lambda_{235}\,^{235}F_T\,^{235}Ue^{\lambda_{235}t_i} + \\ 6\lambda_{232}\,^{232}F_T\,^{232}The^{\lambda_{232}t_i} + \lambda_{147}\,^{147}F_T\,^{147}Sme^{\lambda_{147}t_i}\end{array}\right]}$$

(a10)

and the denominator of the other components also change accordingly. Finally, with $^{235}U$ quantified directly, the overall

uncertainty propagation equation becomes:




$$\sigma_t = \sqrt{\begin{aligned} & \left(\frac{\partial t}{\partial\ ^4He}\sigma_{He}\right)^2 + \left(\frac{\partial t}{\partial^{238}U}\sigma_{238}\right)^2 + \left(\frac{\partial t}{\partial^{235}U}\sigma_{235}\right)^2 + \left(\frac{\partial t}{\partial^{232}Th}\sigma_{232}\right)^2 + \left(\frac{\partial t}{\partial^{147}Sm}\sigma_{147}\right)^2 + \\ & 2\frac{\partial t}{\partial^{238}U}\frac{\partial t}{\partial^{235}U}\sigma_{238-235}^2 + 2\frac{\partial t}{\partial^{238}U}\frac{\partial t}{\partial^{232}Th}\sigma_{238-232}^2 + 2\frac{\partial t}{\partial^{238}U}\frac{\partial t}{\partial^{147}Sm}\sigma_{238-147}^2 + \\ & 2\frac{\partial t}{\partial^{235}U}\frac{\partial t}{\partial^{232}Th}\sigma_{235-232}^2 + 2\frac{\partial t}{\partial^{235}U}\frac{\partial t}{\partial^{147}Sm}\sigma_{235-147}^2 + 2\frac{\partial t}{\partial^{235}Th}\frac{\partial t}{\partial^{147}Sm}\sigma_{232-147}^2 + \\ & \left(\frac{\partial t}{\partial^{238}F_T}\sigma_{Ft238}\right)^2 + \left(\frac{\partial t}{\partial^{235}F_T}\sigma_{Ft235}\right)^2 + \left(\frac{\partial t}{\partial^{232}F_T}\sigma_{Ft232}\right)^2 + \left(\frac{\partial t}{\partial^{147}F_T}\sigma_{Ft147}\right)^2 + \\ & 2\frac{\partial t}{\partial^{238}F_T}\frac{\partial t}{\partial^{235}F_T}\sigma_{Ft238-Ft235}^2 + 2\frac{\partial t}{\partial^{238}F_T}\frac{\partial t}{\partial^{232}F_T}\sigma_{Ft238-Ft232}^2 + \\ & 2\frac{\partial t}{\partial^{238}F_T}\frac{\partial t}{\partial^{147}F_T}\sigma_{Ft238-Ft147}^2 + 2\frac{\partial t}{\partial^{235}F_T}\frac{\partial t}{\partial^{232}F_T}\sigma_{Ft235-Ft232}^2 + \\ & 2\frac{\partial t}{\partial^{235}F_T}\frac{\partial t}{\partial^{147}F_T}\sigma_{Ft235-Ft147}^2 + 2\frac{\partial t}{\partial^{232}F_T}\frac{\partial t}{\partial^{147}F_T}\sigma_{Ft232-Ft147}^2 \end{aligned}}$$

$$(a11)$$

**Appendix B: Implications of Gaussian input uncertainties in HeCalc**

Negative dates are permitted in the probability distributions produced by HeCalc; this is because the input

distributions are presumed to be gaussian, meaning that if the input variables have high relative errors, negative molar amounts

of U, Th, Sm, and He are possible. This behavior is formally correct for gaussian uncertainties, albeit non-physical. For low

count rates associated with high relative uncertainty, a Poisson distribution (rather than gaussian distribution) would be

appropriate, and would prevent negative input values. However, high relative input uncertainties are generally a result of a

measurement being near or below background rather than low count rates where the underlying poisson distribution of the data

is not well approximated by a gaussian. As a result, there are potential instances of negative molar amounts included in the

Monte Carlo calculations.

     In some rare instances when a negative amount of a given parent nuclide is produced in the generation of random

data, the (U-Th)/He date equation may have multiple or no solutions. In these cases, the result is simply removed from the

sample of calculated ages. The total number of such removals is tracked, and if the proportion removed exceeds the requested

precision level, all results associated with the Monte Carlo simulation is reported as NaN (i.e., "not a number") and only the

linear uncertainty propagation results are returned. For typical inputs of routine analyses with a few percent relative uncertainty

(Sect. 5.4), the impact of this phenomenon is entirely negligible.

**Code availability**

715       Version 0.3.3 of the HeCalc software is available at https://zenodo.org/record/6519020. A windows executable

application to run HeCalc is available through the latest release on the software's GitHub repository at

https://github.com/Peter-E-Martin/HeCalc/releases/latest. Code documentation and installation instructions are also available

on the GitHub repository.

**Author contributions**

720       PEM, RMF, and JRM conceptualized the project; JRM curated the data; PEM performed the formal analyses; RMF

and JRM acquired funding; PEM, RMF, and JRM performed the investigation; PEM developed the methodology and wrote

the software; RMF provided supervision; PEM wrote the original draft, and RMF and JRM reviewed and edited the manuscript.

**Competing interests**

The authors declare they have no competing interests.

**Acknowledgements**

We thank Noah McLean for numerous and very helpful discussions while developing HeCalc and writing this paper. HeCalc

and this manuscript were greatly improved following discussions at the 17th International Conference on Thermochronology,

in particular with Danny Stockli, Florian Hoffman, Marissa Tremblay and Kip Hodges. The (U-Th)/He analyses used in the

data compilation presented here were generated by instrumentation funded by National Science Foundation award EAR-

1126991 to Flowers, and awards EAR-1559306 and –1920648 to Flowers and Metcalf.

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
