# Peer review of "Calculation of Uncertainty in the (U-Th)/He System"

_EGUsphere, 2022_

## Author Response (AR1)

**Authors' response to reviewers**

Here we copy the reviews received for this manuscript and respond by noting where changes in the text were made; our original responses to the reviewers contain much high-level discussion that we will not replicate here. We would also like to highlight at a high level that the most important change suggested by each reviewer was a drastic reduction in length, particularly by trimming Section 5. The paper is now substantially shorter (the main text is 17 pages instead of 34), achieved by trimming of the language throughout and by a wholesale revision of Section 5, rewriting and moving much of the original text into appendices. Section 5 now occupies 4.5 pages including figures, instead of 18. The portions of Section 5 that have been moved out of the main text offer important context but we agree with the reviewers that these were not critical to the main point of the paper and again thank the reviewers for noting the potential for this improvement.

**Reviewer 1:**
**This paper discusses the error propagation of (U-Th)/He data. Its introduction claims that "the formal analytical uncertainty in (U-Th)/He dates has never been thoroughly assessed". I have three comments about this statement:**

We have added several sentences in the abstract and introduction to clarify our meaning about the status of published uncertainty calculation methods.

1. **I am sure that the error propagation of the (U-Th)/He method has been worked out before, and probably several times. In fact, I have done so myself, and even implemented it in a publicly accessible computer program: https://ucl.ac.uk/~ucfbpve/heliocalc/.**
2. **One probable reason why nobody published the error propagation formulas for the (U-Th)/He method is the overdispersion that characterises most (U-Th)/He datasets: the scatter of several aliquots from the same samples usually exceeds the precision of the data, by a lot. So, in a sense, the analytical uncertainties are irrelevant. The interplay between analytical uncertainty and overdispersion is discussed by Vermeesch (2010, doi:10.1016/j.chemgeo.2010.01.002), who also covers some aspects of the error propagation problem.**

We have added several sentences in the introduction to more clearly establish that overdispersion, and particularly developing a means of understanding the fundamental causes of overdispersion, are core motivations for publishing this manuscript.

3. **A second reason why error propagation hasn't been discussed much is that it is next to impossible to quantify the analytical uncertainty of the alpha ejection correction, which is one of the main sources of uncertainty in (U-Th)/He dating. Geochronologists slap a nominal uncertainty on this correction, which largely defeats the purpose of rigorous error propagation for the other variables. Unfortunately, the paper under consideration does not address this issue.**

We have added text clarifying the current state of $F_T$ uncertainty quantification and clarifying the language that already existed in the paper. We have also slightly edited our introduction to emphasize this issue as a fundamental motivation for the work.

**Despite these three caveats, I do not object to publishing the error propagation formulas in GChron. However, before this can happen the manuscript needs serious revision. The paper is far too long and can be shortened by at least 50%. I will make some specific suggestions for this later in this review.**
**The paper uses both standard error propagation and Monte Carlo (MC) simulation. I have two comments about this:**

1. **According to the authors, the main advantage of the MC method is its ability to handle skewed error distributions. However, it would be easy to adjust the conventional error propagation to handle the observed skewness. This can be achieved by reformulating the error propagation formula in terms of the log of the variables (e.g., Section 5 of https://doi.org/10.5194/gchron-2-119-2020). Thus, the log of the age could be calculated as a function of the U, Th and He concentrations. An even better solution would be to use log-ratios. See Vermeesch (2010) for details. I am not sure how easy it would be to reformulate the paper and HeCalc code in terms of log(ratio) variables. If the authors find it too difficult, then I guess that the MC approach would be fine as an alternative.**

For now, we have left the code and equations in their original form. Future work could certainly expand on the analytical equations here and provide improvements. As we argue in the paper, even large calculations (several million MC iterations) are performed in a few seconds, fractions of a percent of the time taken to run a (U-Th)/He analysis. Using the MC results to retrieve skewed distributions is fully acceptable in our view.

2. **The actual main advantage of MC error propagation is not mentioned, namely its ability to handle non-Gaussian error distributions. This is particularly pertinent with regards to the alpha-ejection correction (i.e. the uncertainty of the alpha-retention factor Ft). Meesters and Dunai (2002, https://doi.org/10.1016/S0009-2541(01)00423-5) and Hourigan (2005, https://doi.org/10.1016/j.gca.2005.01.024) have shown that compositional zoning can strongly affect the fraction of ejected alpha particles. Matters are further complicated in the presence of broken grains, when the alpha ejection correction may result in overcorrection (Brown et al., 2013, https://doi.org/10.1016/j.gca.2013.05.041). Things are even more difficult for slowly cooled samples, in which alpha-ejection occurs synchronous with diffusive loss of helium. The dispersion caused by all these complexities is difficult to ascertain, but is likely non-Gaussian. The MC approach could be used to explore these effects. I'm not saying that the authors should do this in their paper (because I don't want to make it even longer), but they should at least mention the possibility. Perhaps HeCalc could offer an interface to explore these effects?**

This suggestion is a good one. We have added a sentence highlighting the potential benefits of such an analysis. However, we have also avoided major changes here as this is not critical to the

main motivators for this work. HeCalc is open-source, so any person wishing to implement this feature is more than welcome to use the GitHub interface to do so (either by forking the repository, submitting an issue, or even a pull request on the main branch).

**Detailed comments:**
**Equations 4-10 are unnecessary. They are simply repeating Meesters and Dunai (2005, https://doi.org/10.1029/2004GC000834), and Raphson and Newton (~1711). Incidentally, I do not really see the point of using the Meesters and Dunai (2005) solution as a starting point for a Raphson-Newton algorithm anyway. Their direct solution is accurate to better than 0.1% for ages up to 500Ma, which covers all terrestrial applications of the (U-Th)/He method.**

We have removed these equations and renumbered the equations throughout, excepting the modification to the Meesters and Dunai method, which would be necessary to replicate our work. As argued in the original response, the Newton-Raphson method has very little computational cost, and allows accurate age determinations for any sample, so we have left this in place.

**Equations 11 and 12 could be written more succinctly in matrix form.**
**Equations 14-18 all share the same denominator (which equals df/dt), which could be stored in a variable. All these equations could be put together into a single Jacobian matrix, or moved into the appendix.**

As argued in the original response, the main equations for uncertainty propagation are a major focus of the paper and we feel that they should remain in mostly expanded form in the main text. The Appendix lists additional and fully expanded equations.

**Section 3.3.2 describes a method to choose the optimal number of MC iterations to derive a desired level of precision on the mean value. It just presents the well known "the square root of n" phenomenon, which I think is too trivial a result to occupy so much space (Figure 2 is certainly not necessary). It is also important to note that the square root of n rule only applies to the standard error of the mean. The standard error of the standard deviation (s) is given by s/sqrt(2n-2). I am mentioning this here because the uncertainty of the standard deviation is more relevant than that of the mean, which is never used in the remainder of the paper.**

The algorithm to calculate the number of MC iterations has been revised in the response, and the text deleted/revised accordingly.

**I installed HeCalc on my computer and am happy to confirm that it works. I have not extensively tested it though. I think that the presentation of HeCalc should take greater prominence in the paper. Of course, this will automatically happen if some of the remaining bulk is removed.**

Little of the original HeCalc text has been altered, so it does indeed take a more prominent position in the paper now.

**HeCalc requires that the user provide the uncertainties of the alpha-retention factors 238Ft, 235Ft, 232Ft and 147Ft. However, the paper does not explain how these uncertainties should be obtained. A nominal 5% uncertainty is used in later examples, without proper justification.**

Discussion of the ongoing work to quantify $F_T$ uncertainty has been expanded throughout the paper, including reference to new work by Spencer Zeigler, which provides further justification for the numbers used.

**HeCalc also requires that the user specify the error correlations between the different parameters. However, it does not discuss how to estimate those correlations. Does the CU TRaiL database specify them?**

Error correlations are not explicitly required by HeCalc. We have edited to emphasize this point.

**Minor comment: the paper (and HeCalc) use the awkward convention to report MC uncertainties as "68% confidence intervals". I understand where this comes from: a 1-sigma interval around the mean of a normal distribution covers 68% of that distribution. However, uncertainties are usually reported either as standard errors or as 95% confidence intervals. If the authors want to compare their analytical results with the MC simulations, then a 95% confidence interval would be more elegant.**

We have updated HeCalc to report both 68% and 95% confidence intervals. The relevant sections of the paper have also been updated. However, we have left the 68% confidence interval in place for the discussion.

**Section 5 can be nearly completely removed. The most interesting part of this section is the finding that parent concentrations are a greater contributor to the uncertainty budget than the helium concentrations. This finding could be reported much more succinctly.**

As can be seen in the tracked changes document, large portions of Section 5 that were not critical have been moved into the Appendix. The remaining text has been heavily revised and shortened.

**According to lines 421-423: "when combining uncertainties with equal magnitude, the resulting uncertainty will be only ~1.4 times larger than the input, rather than twice as large as might be expected." Here the authors underestimate the reader. I am certain that the vast majority of geochronologists are familiar with the quadratic addition of uncertainties. Consequently this sentence, as well as the preceding paragraph and Figure 4, can be safely removed.**

We ultimately removed this paragraph; we agree that it contributed little to the original manuscript.

**The paper attributes the reduction of analytical uncertainty with increasing date to the "roll over" of the exponential decay function. This may be correct but is largely irrelevant to real world applications. The observed reduction only expresses itself at >1 Ga, while the**

**vast majority of published (U-Th)/He dates are <200 Ma. At young ages, the helium age equation is linear to a good approximation (https://doi.org/10.1016/j.chemgeo.2008.01.027). Note, however, that the fixed uncertainty of the helium measurements shown in Figure 3 is not realistic: older samples will tend to contain more helium, which can be measured more precisely. This will also cause a reduction of analytical uncertainty, even for Cenozoic samples.**

This subsection has been moved to the appendix.

**In section 5.2, the authors introduce a new definition for skewness. This is a very bad idea. There already exists enough confusion in the geological community about basic statistical concepts. It would be unwise to add to the confusion by redefining widely accepted terms such as skewness. At this point I would like to reiterate the fact that the approximately lognormal uncertainty distribution of the dates could easily be captured analytically by recasting the equations as a function of the log of the age. Simply referring to the percent uncertainty of the age would capture the uncertainty and the skewness with a single number.**

We have not reverted to using the formal definition for skewness. We feel strongly that this numeric value is unintuitive and would provide little benefit to the majority of readers.

**Section 5.4 applies the algorithms to a database of ~3,600 (U-Th)/He dates. It is a shame that this database is not released along with the paper. It must be a treasure trove of useful information! Unfortunately, I don't think that Section 5.4 is particularly interesting. It definitely doesn't deserve seven manuscript pages, four pages and three figures (not counting sub-panels). However, Figure 11 does illustrate my comment at the start of this review effectively: the nominal uncertainty of the alpha-ejection correction dwarfs the other uncertainties, thereby defeating the purpose of the careful error propagation.**

Much of this text has been trimmed, shortening this section substantially. As noted above, we have updated much of the text to emphasize that an outcome of this work is that $F_T$ uncertainty quantification is critical. Recent studies that have quantified geometric uncertainty in $F_T$ now enable this uncertainty to be propagated.

As noted in our previous response, much of the data in the compilation comes from samples that CU TRaIL was contracted to run. We do not "own" these data and they therefore are not ours to release outside of these anonymized and derived figures. Much of the data produced by CU TRaIL for internal research projects are published or are in the process of being published, and are easily discoverable in the literature.

**Lines 615-616: "a challenge to interpreting data with asymmetrical uncertainties is that no widely used inverse thermal history modeling software for (U-Th)/He data permits the input of asymmetrical uncertainty" I'm not sure how HeFTy handles the analytical uncertainty of (U-Th)/He data, but if I seem to recall that QTQt essentially inflates the uncertainties until they account for the overdispersion of the data. This means that the uncertainties are, effectively, ignored. HeFTy probably does something similar, because otherwise its formalised hypothesis tests would fail. Ideally, thermal history inversions**

**should aim to predict the uncorrected (U-Th)/He dates, ignoring the alpha ejection correction. As mentioned before, this is because alpha ejection occurs concurrently with thermal diffusion. So it is not a constant but a variable that depends on the thermal history (Meesters and Dunai, 2002).**

No change to the text appears required in response to this comment.

**Equations a1-a10 all have the same denominator. Storing this denominator in a variable would avoid a lot of duplicate text. You could then even put all these equations into a single concise Jacobian.**

Appendix A has the express purpose of providing equations in their expanded form, such that it is possible to enter them directly into spreadsheet programs. We therefore prefer not to alter them, or to convert them to matrix form.

**I apologise if this review comes across as overly critical. I think that this paper (and the HeCalc program) could serve a useful purpose. My opinions is that it would be greatly improved by trimming it down to the important parts. Perhaps the paper could be recast as one of GChron's popular "Technical notes"? This would provide a nice way to present HeCalc to the world, whilst reviewing the error propagation problem.**

**Reviewer 2:**
**This manuscript presents an algorithm for determining U-Th-Sm/He dates in accessory minerals and then derives uncertainty propagation equations for them, combining uncertainties in measurements of nuclide amounts and geometric correction factors. They also introduce a Monte Carlo method and compare the two. These algorithms are implemented in a python package hosted on zenodo.**
**A consistent set of uncertainty propagation equations for the community is a welcome addition to the literature, although it is unlikely that it will strongly affect science derived from the measurements due to pervasive overdispersion.**

**I've had the opportunity to read the review by Pieter Vermeesch prior to writing this, and that saves me a lot of time writing my review because I came to basically the same conclusions as him. I don't think it's necessary for me to state specifically on which points I agree, but one that affects the manuscript substantially is the suggestion to cut down the length. I agree that section five can be completely removed. I don't think it's necessary for the authors to include log functions to the analytical solutions, but I agree that it would be a superior technique.**

As noted previously, the manuscript has been substantially edited for brevity. In particular, large chunks of Section 5 have been rewritten, revised, and/or moved to the appendices. The end result is that Section 5 is now only 4.5 pages, including figures.

**What I think would be very useful is a comparison of before/after of typical analyses from the authors lab. A few representative datasets would be just fine. I think this would be of great value in showing readers whether this approach changes assigned uncertainties in any substantial way. Maybe it doesn't, but that's not necessarily a bad thing.**

We have not revised the manuscript to include this suggestion because we do not think specific examples from CU TRaIL would be broadly applicable. We have, however, made several changes to highlight that developing a unified method of uncertainty propagation would allow inter-laboratory uncertainty comparisons.

**I think it's a missed opportunity somewhat to not assess the accuracy of assigned uncertainties in the U-Th-Sm, and the He analyses. I appreciate that the authors have carved the manuscript such that this is "out of scope" (which is their prerogative), but the brief comment that tracers are often added by pipetting (without discussing the implications of adding a spike isotope using a technique with such a high variability) suggests to me that the radionuclide measurements may be underestimated. Not to mention that the treatment of under/over spiking and blank subtraction (to name a couple), in my experience, are dealt with in a highly variable way by different people in the U-Th-Sm/He community. I guess that leaves space for the next paper!**

We agree that this is beyond the scope of the current paper; we have not made changes to include this, but it would be nice to see in the literature in the future.

**I have a few shorter comments below:**
**L9: What quantities are $^4$He and "radionuclide" here. Are they amounts, concentrations….? Please briefly define $F_T$ in the abstract for a non-specialist, particularly because it's referred to as particularly important later in the abstract.**

We have added clarification in the abstract for both of these points.

**L11: Is this relative or absolute uncertainty?**

We now define this explicitly in the abstract.

**L15: Again, is this concentration?**

This is also now clearly defined.

**L15: What is the confidence level for these estimates? 95%? 2sigma?**

We have added a specification that this is at the equivalent 1-sigma level (68% confidence interval)

**L34: I assume that by "kinetic" the authors mean diffusion kinetics.**

Yes, we have clarified this.

**L76: "ppm" is, in general, ambiguous and best practice is that it should be avoided. I appreciate that there is an implicit convention in some geochemical subfields that it refers only to µg/g, but it is not always the case and there is no disadvantage to being explicit and using the SI-consistent µg/g.**

We have changed this phrasing throughout the manuscript.

**L80: "sector" should say "magnetic sector"**

This has been added.

**L81: The technique is called "isotope dilution" not "isotope spike".**

This has been changed.

**L82: I'm not sure what "ratioed mass spectrometric measurements" are?**

We have added clarifying text that we are referring to measured isotope ratios.

**L103-104: It's not clear to me that this is true. For example, if a mixed U-Th-Sm tracer is used and the mass of spike solution is relatively small (which is usually the case), the uncertainty in the amount of spike added (which propagates directly onto the amount of U, Th, and Sm calculated) will be relatively large and since the tracer is added as a mixture, that component will be highly correlated. It takes special care to get weighing errors to less than 1%, and with small amounts of spike they can easily be in the 5-10% range. When pipetting without weighing (which is what is implied here) the problem can be much worse. Pipetting consistency can vary, for sure, but for 25 µl the relative standard deviation on masses dispensed can be 3-5%, which translates directly to a 3-5% uncertainty on the measured quantities. This seems like it should be large enough to matter?**

We have performed some in-house pipette repeatability testing and reassured ourselves that for CU TRaIL, this is not relevant. However, it certainly could be for other labs, so we have edited the text to make it more open-ended because this needs to be evaluated on a per-lab basis.

**L117: 2-9% in what? The Ft correction or the final date? And is this a bias (e.g., the technical definition of "error") or additional uncertainty on the date?**

We have added clarification that this refers to uncertainty in $F_T$ value.

**L131: Here and elsewhere the word "variance" is used. It's unclear as to whether this is referring to a moment of the gaussian distribution or as a casual synonym for uncertainty or data scatter.**

We replaced the word "variance" with "dispersion" to mean data scatter.

**L366: A Th/U (by mass) ratio of 1.25 is \*not\* typical of zircon, it is unusually high. Looking at the georoc database of zircon compositions, after doing some data culling, the median value is about 0.6 (mean = 0.8). That sounds much more realistic to me than 1.25.**

Thanks again for catching this. It is now corrected throughout.

**L420: This statement should be justified or referenced, or else removed. It sounds a little bit like it is underestimating the math and stats skills of typical geochronologists.**

We have removed this paragraph.

---

## Author Response (AR2)

1. I really don't think it is necessary to explain well-known concepts such as the Newton-Raphson method (Section 3.1) and Monte Carlo error propagation (Section 3.3) in such minute detail. Thus, I think that Equations 4-6 and Figure 1 can be safely removed from the paper.

We have not made any changes here. While we respect the editor's request to shorten the paper where practicable, we know through experience (as argued from the initial responses) that there are practitioners and users of the (U-Th)/He community for whom these topics are not second nature and would benefit from the inclusion of these topics. We therefore strongly feel that to make this topic as accessible as possible, these background portions of the paper are important to retain.

2. Equations 7, 8, 19 and a11 are wrong. The covariances should not be squared.

This is an embarrassing mistake primarily caused by the lack of a generally accepted symbol for variance (i.e. $\sigma_a$ would typically be the standard deviation of $a$, while $\sigma_{ab}^2$, rather than $\sigma_{ab}$, would be the covariance of $a$ and $b$). We appreciate that prof. Vermeesch caught it before the paper went to press. We have inspected the code and fortunately, the correlated errors are calculated correctly by HeCalc. Rather than changing the equations, we have corrected the definitions (now correctly identifying $\sigma_{ab}^2$ as the covariance rather than describing $\sigma_{ab}$ as the covariance). This notation is in alignment with McLean et al., 2011 and will hopefully be most familiar to the general geochronology community.

3. I was disappointed that the revision retains the ad-hoc definition of skewness (or "skew"). Not only is this definition confusing, but it is also unnecessary. As I mentioned in my first review, the skewed error distributions likely fit a simple lognormal distribution. If this is the case, then the skewness would disappear after taking logs, and the analytical uncertainty could be adequately captured with a single number, namely the standard error of the log. The absolute standard error of a logarithmic quantity corresponds to the relative standard error of its linear equivalent. Had the authors carried out their linear error propagation in log space, then this would have avoided the need for Monte Carlo simulations. Since two thirds of the paper is dedicated to explaining the difference between the linear and Monte Carlo uncertainty estimates, this means that two thirds of the paper is essentially unnecessary. This includes Figure 3, 4, 5, C2 and the entire appendices D and E. Taking logs would also avoid the issue with negative dates that is mentioned in Appendix B.

To avoid the confusion that the reviewer points out, we have added in the formal skewness values to the text, table, and plots.

On the topic of linear vs log-space equations, prof. Vermeesch acknowledged previously that the Monte Carlo approach achieves the same results, and is sufficiently fast that no real efficiency is sacrificed. On the other hand, we know through experience and anecdote that many labs are using linear (not log) approximations for their uncertainty propagation. The reason for the inclusion of linear equations here is to 1) formalize the process for those already using these approximations, and 2) provide a comparison point for those labs to decide for themselves whether the age equation non-linearity causes sufficient error to move away from this approximation. In the CU TRaIL, where dates >1 Ga are measured, this is a relevant factor. In

prof. Vermeesch's lab, where such ages are not observed, any version of uncertainty propagation is probably fine to a very good approximation.

Other comments:

1. Section 1 claims that the uncertainty of the alpha-retention factor Ft is "increasingly well-constrained". This is only the case for the measurement of the grain dimensions. Unfortunately, there has been little or no progress in quantifying the larger effect of compositional zoning (Hourigan et al., 2005). This effect is, essentially, unknowable.
It is untrue that the effects of compositional zoning are unknowable. Compositional zonation is quantifiable via published LA-ICP-MS mapping (Farley et al., 2011) and drilling (Johnstone et al., 2013) methods. We now cite these papers in the manuscript. Whether it is worthwhile to routinely acquire such data for typical studies is another question.

2. Section 3.3. typo "6odelling"
Corrected.

3. Section 3 cites Efron and Tibshirani (1986) as a reference for Monte Carlo error propagation. However, this paper discusses bootstrapping, which is a different procedure than the one described in the manuscript.
This is a holdover from a much older version of the paper. This reference has been removed.

4. Table 1. Typo: "convariance"
Corrected.

5. Section 4.2 defines a 95% Monte Carlo confidence interval as the difference between the 2.275 and 97.725 percentiles. However, 97.725 - 2.275 = 95.45 ≠ 95!
This is another good catch; we had simply converted from a 2-sigma equivalent. This is now corrected in both the paper and in the code.

6. The denominators of Equations a1-a8 are all the same. It would save lots of ink if they were replaced by a separate parameter. I suggest 'd' for 'denominator'.
In the interest of shortening the paper, we have removed these equations entirely as they are duplicates of equations in the main text. The denominator of the remaining equations for direct quantification have been updated with the summation notation used in the main text.

7. Figure C1: the three columns of this plot are, essentially, the same (with the first and third being exactly the same). This reflects the fact that U, Th, He and Ft are simple multiplicative factors in the age equation. So the sensitivity of the age is, essentially, the same for them. I don't think that this figure is very informative. I suggest that it be removed.
We appreciate that this figure isn't very informative for Prof Vermeesch given his statistical expertise, but we feel that it will be informative for our target audience of method practitioners who have less statistical background. Just as in the first round of edits, we mainly wish to convey an intuitive sense of which components of uncertainty are most important for uncertainty in the

(U-Th)/He date. In a paper focused on explaining uncertainty sources and how they propagate, this figure should be included.

8. Figure C3 is misleading. As explained in its caption, it exaggerates the non-linearity of the (U-Th)/He age equation by decreasing the uranium decay constant. In other words: the non-linearity is a straw man problem. It only affects >1Ga samples, which are never found on Earth anyway. Consequently, I think that Appendix C can be removed as well.

Ages >1 Ga absolutely are found, and measured, on Earth. They are routinely observed in CU TRaIL. For that reason, we feel strongly that appendix C should be retained. However, the detailed text explaining the rollover phenomenon is likely to be review for those with advanced statistical knowledge and uninteresting to others. We have therefore removed the final paragraph and figure of appendix C in the interest of shortening the paper.

---

## Author Response (AR3)

There are only a few comments from the head editor to respond to in this accepted round of reviews. We respond in-line with blue text below.

I agree with the associate editor that this is suitable for publication. I have a few final comments:

-- This could be published either as a 'research article' or a 'technical note.' There is no size limit on a 'technical note', although this is confusing on the Gchron website because 'technical notes' and 'short communications' are largely described together. Personally, I would lean toward 'technical note' because for the most part this paper reviews data and mathematical methods that are already known, but the authors should use their best judgement. Please note that 'technical notes' at GChron have exactly the same review and bibliographic treatment as 'research articles' -- they are not considered a less rigorous category. If the authors wish to publish this as a 'technical note,' please contact the Copernicus editorial staff with a copy to myself and the AE.

We appreciate being given the option here. Our preference is to leave this article as a "research article".

-- Please carefully check all the equations in the paper during typesetting and production.

We have checked the equations again prior to this submission and found no errors. We will also check again during the formatting process.

-- One theme of the reviews and responses for this paper is the tension between providing a mathematically concise description of the error propagation scheme and providing a description that can easily be understood by students or others with a less extensive mathematical background. In this paper the authors have argued in favor of the latter, which is totally reasonable in light of the intended purpose of the paper. However, a potential problem with this approach, that appears to be recognized but not explicitly discussed in the correspondence, is that the authors of this paper are, at least implicitly, proposing that the approach described in the paper should be adopted as a standard in the field. This leads to the risk that a more mathematically concise error propagation scheme (e.g., calculation of uncertainties in log rather than by Monte Carlo), even though it is also correct, might not be accepted in future publications because it would be viewed as noncompliant with the more simplistic standard proposed here. However, (i) the authors have been very good about keeping prescriptive remarks and recommendations out of the paper (this is good), and (ii) the purpose of this paper is, as articulated by the authors, to explain the method to students. Thus, I don't think any action is needed to address this.

This is a nice observation of one of the main points raised in review that we don't think was ever stated explicitly. We did notice that the first line of the abstract had the potential to seem as if we wished to be prescriptive in this method. We have therefore slightly altered (and shortened) the first line to read: "Although rigorous uncertainty reporting on (U-Th)/He dates is key for interpreting the expected distribution of dates within individual samples and for comparing dates generated by different labs, the methods and formulae for calculating single-grain uncertainty have never been fully described and published."